# Structure of the human outer kinetochore KMN network complex

**Stanislau Yatskevich** [1,2] ✉, **Jing Yang** [1], **Dom Bellini** [1], **Ziguo Zhang** [1] **& David Barford** [1] ✉

Faithful chromosome segregation requires robust, load-bearing attachments of chromosomes to the mitotic spindle, a function accomplished by large macromolecular complexes termed kinetochores. In most eukaryotes, the constitutive centromere-associated network (CCAN) complex of the inner kinetochore recruits to centromeres the ten-subunit outer kinetochore KMN network that comprises the KNL1C, MIS12C and NDC80C complexes. The KMN network directly attaches CCAN to microtubules through MIS12C and NDC80C. Here, we determined a high-resolution cryo-EM structure of the human KMN network. This showed an intricate and extensive assembly of KMN subunits, with the central MIS12C forming rigid interfaces with NDC80C and KNL1C, augmented by multiple peptidic inter-subunit connections. We also observed that unphosphorylated MIS12C exists in an auto-inhibited state that suppresses its capacity to interact with CCAN. Ser100 and Ser109 of the N-terminal segment of the MIS12C subunit Dsn1, two key targets of Aurora B kinase, directly stabilize this auto-inhibition. Our study indicates how selectively relieving this auto-inhibition through Ser100 and Ser109 phosphorylation might restrict outer kinetochore assembly to functional centromeres during cell division.

Kinetochores are large macromolecular complexes that couple the forces of microtubule depolymerization to chromosome movement during mitosis and meiosis[1–3]. The chromatin-proximal inner kinetochore, also known as the CCAN, assembles specifically at the centromere by recognizing the centromere-specific histone H3 variant CENP-A. Inner kinetochores, together with CENP-A nucleosomes (CENP-A[Nuc]), form load-bearing attachments at centromeric chromatin and recruit the outer kinetochore, which directly couples CCAN to microtubules in the mitotic spindle. Although CCAN is present at the centromere throughout the cell cycle, in metazoans the outer kinetochore is recruited to centromeres only during mitosis[4]. The outer kinetochore also scaffolds the spindle assembly checkpoint (SAC), a critical signaling pathway that ensures accurate chromosome segregation[5–7].

Whereas the inner kinetochore diverged throughout evolution, the outer kinetochore is compositionally and structurally conserved in most species and is the primary force-coupling device between chromosomes and the mitotic spindle[8]. The outer kinetochore is made up of the central and evolutionarily conserved ten-subunit KMN network complex, comprising the MIS12 (Mis12, Dsn1, Nsl1, Pmf1), NDC80 (Ndc80, Nuf2, Spc24, Spc25) and KNL1 (Knl1, ZWINT) complexes as well as auxiliary microtubule-associated components, such as the SKA complex, that vary between species. The rod-shaped MIS12C protein is a central interaction hub, directly binding the RING finger, WD repeat, DEAD-like helicase (RWD) domains of Spc24–Spc25 (Spc24–Spc25[RWD]) of NDC80C, as well as the Knl1 RWD domains (Knl1[RWD]) of KNL1C[9–12], thereby coordinating the microtubule-binding (NDC80C) and checkpoint signaling modules (KNL1C) of the KMN network. MIS12C is one of the primary KMN network linkages to the centromere. At the bottom of the MIS12C stalk, two head domains interact with linear peptide motifs of the inner kinetochore components—CENP-C and CENP-T[9,10,13–19]. Crystallographic studies of human and yeast CENP-C interactions with

[1]MRC Laboratory of Molecular Biology, Francis Crick Avenue, Cambridge, UK. [2]Present address: Genentech, South San Francisco, CA, USA. ✉e-mail: yatskevs@gene.com; dbarford@mrc-lmb.cam.ac.uk

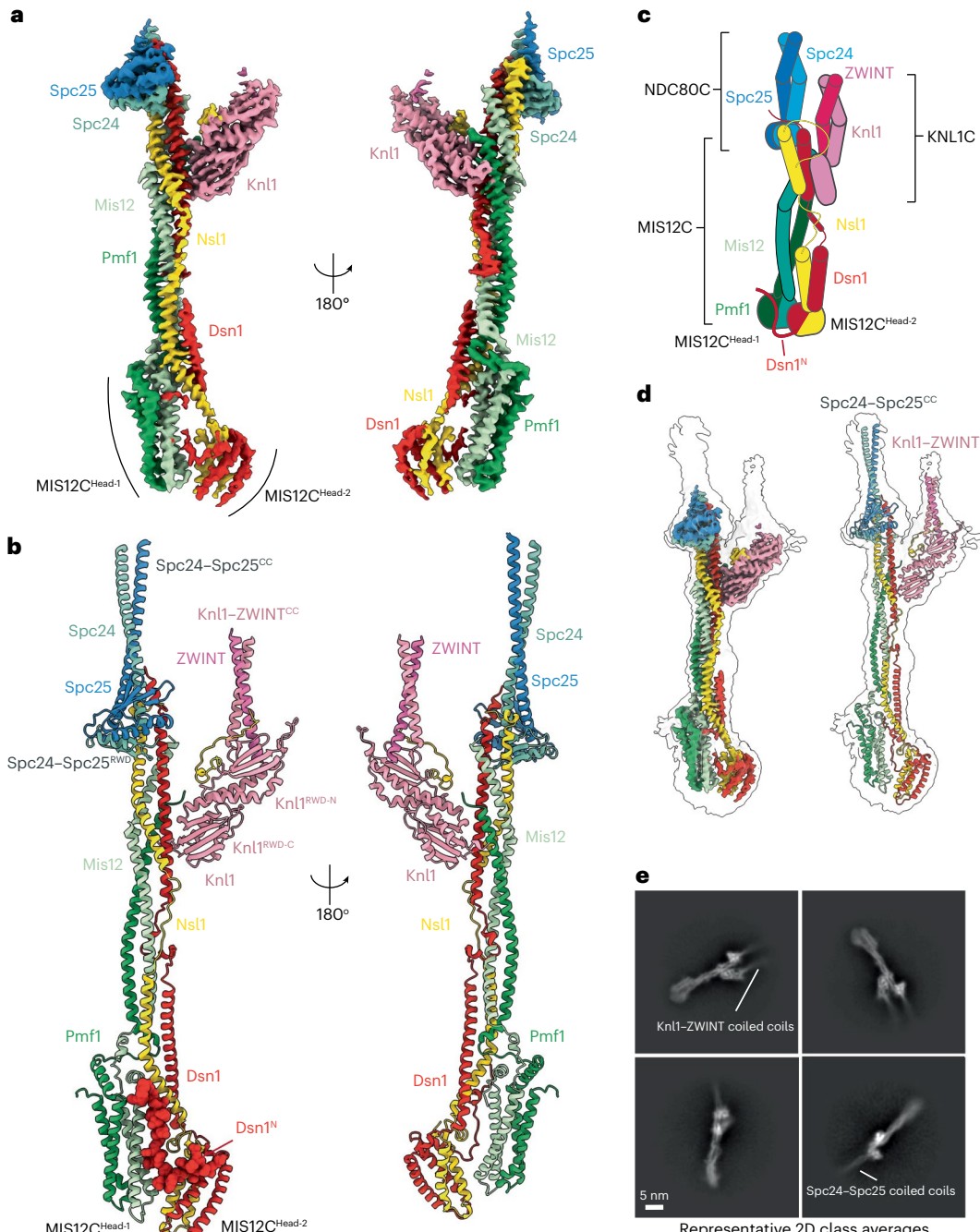

**Fig. 1 | Overall architecture of the human KMN$^{Junction}$ complex. a,** Composite cryo-EM density map of the human KMN$^{Junction}$ complex, composed of the rigid Spc24–Spc25–MIS12C–Knl1–ZWINT body derived from the Body 1 (Extended Data Fig. 2c) and the more mobile MIS12C$^{Head-1}$–MIS12C$^{Head-2}$ body derived from the Body 2 (Extended Data Fig. 2c). **b,** Molecular model of the human KMN$^{Junction}$ complex, highlighting the auto-inhibitory Dsn1$^N$ region in space-filling representation. Spc24–Spc25$^{CC}$ and Knl1–ZWINT$^{CC}$ coiled coils are shown. Coordinates traced: G86–W197 (Spc24), D80–N224 (Spc25), R2097–H2342 (Knl1), Q183–G221 (ZWINT). **c,** Schematic of the human KMN$^{Junction}$ complex, with diagrams of the peptide linkages present in the complex. **d,** A high-resolution sharpened composite map (left) and molecular model (right) of the human KMN$^{Junction}$ complex fitted into the transparent gray consensus unmasked map of the KMN network complex that shows coiled-coil densities of the Spc24–Spc25 and Knl1–ZWINT complexes. **e,** Representative cryo-EM 2D class averages of the KMN network showing coiled coils of Spc24–Spc25 and Knl1–ZWINT complexes projecting from the central KMN$^{Junction}$ complex.

MIS12C have shown that the 45 amino-terminal residues of CENP-C contact the head domain formed by Mis12 and Pmf1 (MIS12C$^{Head-1}$), whereas the head domain of Dsn1 and Nsl1 (MIS12C$^{Head-2}$) projects laterally relative to MIS12C$^{Head-1}$. CENP-T binds MIS12C in a manner that competes with CENP-C binding and requires CENP-T phosphorylation by CDK1, suggesting that CENP-C and CENP-T bind a similar region of MIS12C, but the molecular details of this interaction are not understood[17,18].

NDC80C is the primary and essential force-coupling component of the kinetochore in nearly all eukaryotes[20]. NDC80C contains an approximately 50-nm-long coiled coil, which separates the microtubule-binding calponin homology (CH) domains of Ndc80–Nuf2 at one end from Spc24–Spc25 interactions with MIS12C at the other[21–24]. Multimerization of NDC80C allows it to bind and track both growing and depolymerizing microtubules[20,25], a property that is dependent on a conserved coiled-coil loop (Ndc80$^{Loop}$) that introduces a break

into a continuous Ndc80–Nuf2 coiled coil[26,27]. The CCAN component CENP-T also recruits two copies of NDC80C in human cells, independently of the MIS12C pathway, by forming peptidic interactions with Spc24–Spc25[RWD] (refs. 17,28,29).

Human Knl1 is a 2,342-residue protein that is predicted to be mostly unstructured. The only folded region of Knl1 is located at its carboxy terminus, comprising Knl1[RWD] together with a predicted coiled-coil element, with Knl1[RWD] being sufficient for Knl1 localization to kinetochores[12,30]. The coiled-coil region of Knl1 is necessary to recruit its constitutive binding partner ZWINT[12]. The major function of Knl1 is to orchestrate SAC signaling through the recruitment of key SAC components, such as Bub1, BubR1 and Bub3, using numerous conserved motifs[5]. ZWINT is also responsible for recruitment of the ROD–Zwilch–ZW10 (RZZ) complex, which forms the fibrous corona around unattached kinetochores and facilitates SAC signaling[31]. The extreme N-terminal region of Knl1 has been shown to have an affinity for microtubules, suggesting that Knl1 also contributes to the formation of kinetochore–microtubule attachments[32,33].

Negative-stain electron microscopy (EM) and rotary shadowing EM elucidated the global architecture of the KMN components as an elongated rod-like complex[10,17], yet the molecular details and the functional significance of how the constituent KMN complexes interact have remained elusive. Additionally, multiple studies suggest that MIS12C requires activation by Aurora B phosphorylation. Inhibiting Aurora B kinase activity disassembled kinetochores in Xenopus extracts[34]. In human cells, mutation of the two key Aurora B target residues, Ser100 and Ser109, located within an N-terminal segment of Dsn1 (Dsn1[N]), importantly impairs Dsn1 localization to kinetochores and compromises assembly of the functional outer kinetochore[35,36]. Alanine substitution of the equivalent Ser100 and Ser109 residues in yeast is lethal and affects MIS12C recruitment to inner kinetochores, highlighting the functional importance and evolutionary conservation of these residues[37]. Biochemical studies indicated that Dsn1[N] auto-inhibits the interaction of MIS12C with both CENP-C[9,10,35] and CENP-T[18], and that Aurora B kinase phosphorylation relieves this inhibition. However, the structural basis for Dsn1[N]-mediated auto-inhibition, and how this is relieved by phosphorylation to activate binding of MIS12C to the inner kinetochore, is not understood.

To address these questions, we reconstituted the human KMN network and determined its structure using cryogenic EM (cryo-EM). We observed a rigid prong-shaped structure formed by NDC80C–MIS12C–KNL1C. Their mutual interactions are mediated by moderately sized protein-protein interfaces that are further stabilized by extensive peptide linkages, generating a configuration in which NDC80C and KNL1C extend perfectly parallel to one another from the central MIS12C scaffold. We also observed that MIS12C exists in an auto-inhibited state, with MIS12C[Head-1] and MIS12C[Head-2] positioned closely beside one another. Dsn1[N] directly stabilizes the interaction of the two MIS12C head domains, with Ser100 and Ser109 mediating auto-inhibiting interactions. In this binding mode, Dsn1[N] competes with CENP-C, and likely CENP-T, for the CCAN-binding interface of MIS12C, explaining how recruitment of KMN to the inner kinetochore might be regulated. We demonstrate that Aurora B kinase phosphorylation would release the Dsn1[N]-mediated auto-inhibition, thereby regulating MIS12C–CENP-C interactions.

## Results

### Overall architecture of the KMN network junction

We purified the three complexes of the KMN network; all proteins, apart from Knl1, were full length (Extended Data Fig. 1a). For this study, we used an extended version of the Knl1 C terminus comprising residues 1870–2342, which is sufficient for kinetochore localization and ZWINT binding[12]. Our Knl1 construct also contained all regions of Knl1 predicted by AlphaFold2 to be folded[38] (Extended Data Fig. 1b). We reconstituted the complete KMN network using size-exclusion

**Table 1 | Cryo-EM data collection, refinement and validation statistics**

|  | KMN network complex (EMDB-17814, PDB 8PPR) |
|---|---|
| **Data collection and processing** |  |
| Magnification | ×81,000 |
| Voltage (kV) | 300 |
| Electron exposure (e$^-$/Å$^2$) | 45 |
| Defocus range (µm) | 1.2–2.4 |
| Pixel size (Å) | 1.059 |
| Symmetry imposed | $C_1$ |
| Initial particle images (no.) | 2,549,620 |
| Final particle images (no.) | 599,831 |
| Map resolution (Å) | 3.0 |
| FSC threshold | 0.143 |
| Map resolution range (Å) | 2.7–20 |
| **Refinement** |  |
| Initial model used | AlphaFold2 |
| Model resolution (Å) | 2.7–3.8 |
| FSC threshold | 0.143 |
| Model resolution range (Å) | 2.4–3.2 |
| Map sharpening $B$ factor (Å$^2$) | −61.68 |
| Model composition |  |
| Non-hydrogen atoms | 11,354 |
| Protein residues | 1,388 |
| Ligands | – |
| $B$ factors (Å$^2$) |  |
| Protein | 126.08 |
| Ligand | – |
| R.m.s. deviations |  |
| Bond lengths (Å) | 0.003 |
| Bond angles (°) | 0.567 |
| Validation |  |
| MolProbity score | 1.64 |
| Clashscore | 12.5 |
| Poor rotamers (%) | 0 |
| Ramachandran plot |  |
| Favored (%) | 97.88 |
| Allowed (%) | 2.04 |
| Disallowed (%) | 0.07 |

chromatography (SEC), with all components co-eluting (Extended Data Fig. 1c). We subjected the reconstituted KMN network to cryo-EM analysis and obtained a 3.0-Å-resolution reconstruction of the KMN network junction (KMN[Junction]) containing complete MIS12C, Spc24–Spc25 and all folded domains of KNL1C (Fig. 1, Extended Data Fig. 2a–e and Table 1). The only components that we did not observe at high resolution were the Ndc80 and Nuf2 proteins, which seem to be flexibly connected to Spc24–Spc25 at the tetramerization junction.

At the core of the KMN[Junction] complex is MIS12C, which comprises a tetrameric coiled coil connected to two head domains (MIS12C[Head-1] and MIS12C[Head-2]) (Fig. 1a–c and Supplementary Video 1). Compared with their positions in a previously reported MIS12C crystal structure[10], MIS12C[Head-1] and MIS12C[Head-2] in our reconstruction are closely aligned,

an interaction stabilized by Dsn1[N], which folds along the MIS12C[Head-1] surface (Fig. 1b). This generates an auto-inhibited MIS12C state with an occluded CENP-C binding site, described in detail below. The Spc24–Spc25[RWD] and Knl1[RWD] domains are rigidly docked onto the top of the MIS12C stalk (Fig. 1b), and their interactions with MIS12C are further stabilized by extensive peptide interactions with the C termini of the MIS12C subunits Dsn1 and Nsl1. ZWINT and Knl1 form a coiled coil immediately N-terminal to the Knl1[RWD-N] domain, consistent with this segment of Knl1 being required for ZWINT interactions in vivo[11]. Overall, our structure has a similar global shape to the previously reported negative-stain reconstruction of the truncated KMN complex[11], and we did not observe any direct protein-protein interactions between NDC80C and KNL1C.

A salient feature of KMN[Junction] is the coiled coils of Spc24–Spc25 and Knl1–ZWINT that run perfectly parallel to one another, giving the entire assembly a prong-like shape (Fig. 1b and Extended Data Fig. 3a). The coiled-coil arrangement projects the microtubule-binding Ndc80–Nuf2 and N terminus of Knl1 in the same direction and away from the MIS12C head domains that associate with the inner kinetochore. Therefore, a rigid association of MIS12C–RWD coiled coils results in a long, rigid structure that interacts with the inner kinetochore at one end and microtubules of the mitotic spindle at the other, defining a precise polarity between the centromere-proximal MIS12C[Heads] and microtubule-proximal Ndc80–Nuf2 coiled coils. Although the resolution of the Spc24–Spc25 coiled coils markedly decreases distal to the central MIS12C stalk, the coiled coils are apparent at a lower cryo-EM density threshold and in two-dimensional (2D) class averages, allowing an important extent of the Spc24–Spc25 coiled coils to be traced (Fig. 1d,e and Extended Data Fig. 3a).

## Dsn1 N terminus stabilizes the MIS12C auto-inhibited state

In our reconstruction, MIS12C[Head-2] directly contacts MIS12C[Head-1], resulting in an auto-inhibited MIS12C state (Fig. 2a,b and Extended Data Fig. 3b). This state is stabilized by the Dsn1[N] auto-inhibitory segment (residues 93–113), which forms two contacts with MIS12C[Head-1]: (1) a linchpin formed by the WRR motif (residues 96–98) of Dsn1[N], which locks into the hydrophobic pocket at the Mis12–Pmf1 interface and connects Mis12–Pmf1 to the coiled-coil connector-α3 helices of Dsn1–Nsl1 (Fig. 2a–c); and (2) a zipper-like interaction of the RKSL motif (residues 107–110) that bridges MIS12C[Head-1] and MIS12C[Head-2] through both electrostatic and non-polar interactions (Fig. 2a,b,d and Extended Data Fig. 3b). The WRR linchpin is stabilized by a short α-helix immediately to its C terminus, with Ser100 forming interactions with the peptide backbone of the linchpin and holding it in place (Fig. 2c). We also observe a direct interaction between the tips of Nsl1 and Mis12 across the two heads, stabilizing the auto-inhibited state (Fig. 2c).

The cumulative outcome of these interactions is that the two MIS12C head domains are closely associated, with little accessible space between them. This effectively blocks access to the N-terminal half of the CENP-C binding interface on MIS12C[Head-1] (Fig. 2e). Overlaying the MIS12C–CENP-C crystal structure[10] onto our auto-inhibited MIS12C structure, we observe a direct clash between CENP-C and Dsn1[N] in addition to the steric clash of both MIS12C[Head-2] and the Dsn1–Nsl1 connector-α3 helices with parts of CENP-C (Fig. 2e). The N termini of the Dsn1–Nsl1 connector-α3 helices undergo a modest bend during the transition from the MIS12C auto-inhibited state to the MIS12C–CENP-C complex state; the bend contributes to creation of the CENP-C-binding site (Fig. 2e,f). Overall, in the MIS12C auto-inhibited state, CENP-C residues 6–22 (CENP-C (N); Fig. 2e) are precluded from binding to MIS12C[Head-1], whereas CENP-C residues 28–48 (CENP-C (C); Fig. 2e) are not prevented from interacting with the back of MIS12C[Head-1] (Extended Data Fig. 3c,d). Thus, although MIS12C auto-inhibition does not completely eliminate the interaction between CENP-C and MIS12C, it importantly reduces their binding affinity. Complete CENP-C binding would require disengagement of Dsn1[N] and opening of the MIS12C

central channel by repositioning MIS12C[Head-2] outwards, as observed in the MIS12C–CENP-C crystal structure[10]. This relieves steric clashes of MIS12C[Head-2] with CENP-C residues His8 to Asn11 and Arg14 to Phe17, and concomitantly positions Asp105, Glu112 and Asp113 of the Nsl1 connector-α3 helix to form hydrogen bonds with Arg14 and Arg15 of CENP-C (Fig. 2f). In the MIS12C–CENP-C complex[10], the guanidinium group of CENP-C Arg15 occupies a similar position to that of Arg98 of Dsn1[N] in the MIS12C auto-inhibited structure (Fig. 2f). Additionally, the shifted Dsn1–Nsl1 connector-α3 helices optimally position a non-polar pocket to engage the aromatic side chain of CENP-C Phe17 (Fig. 2f). Strikingly, auto-inhibition through Dsn1[N] is stabilized by two key targets of the Aurora B kinase[16,32,35,36]: Ser100 stabilizes the WRR linchpin (Fig. 2c), and Ser109 stabilizes the RKSL zipper (Fig. 2d). Our modeling suggests that phosphorylation of either of these residues would importantly destabilize Dsn1[N] and alleviate MIS12C auto-inhibition.

We used isothermal titration calorimetry (ITC) to test this structural model by measuring the affinities of MIS12C for both a peptide modeled on CENP-C and a Dsn1[N] auto-inhibitory peptide, including structure-guided interface mutants (Fig. 2g and Extended Data Fig. 4). We observed that the N-terminal region of CENP-C (residues 2–22; CENP-C[2–22]), whose binding site on MIS12C is sterically occluded in the auto-inhibited state, bound to MIS12C with an affinity of 485 nM, a value similar to previous fluorescence-polarization-based measurements[10,13]. Deletion of Dsn1[N] (residues 6–113; MIS12C[Dsn1ΔN]), including the Dsn1[N] auto-inhibitory segment, enhanced CENP-C[2–22] binding to MIS12C by 50-fold, to 10.2 nM, indicative of a tight interaction. Introducing phosphomimetic substitutions at the key regulatory Aurora B phosphorylation sites of Dsn1[N] (S100D and S109D; MIS12C[Dsn1-S100D S109D]) resulted in a 25-fold increase in MIS12C affinity for CENP-C[2–22] (dissociation constant ($K_d$) of 20.4 nM; Fig. 2g and Extended Data Fig. 4c). Overall, our results agree with prior data that deleting a ten-residue segment of Dsn1 incorporating Ser100 and Ser109 increased the affinity of MIS12C for residues 1–21 of CENP-C by 90-fold[10], although a smaller increase in CENP-C binding to a phosphomimetic mutant of MIS12C has also been reported[35]. Consistent with our in vitro results, substituting Ser100 and Ser109 with Ala, which prevents their phosphorylation, importantly reduced Dsn1 localization to kinetochores in human cells, whereas introducing equivalent substitutions in yeast results in loss of viability[35–37], suggesting that auto-inhibition is a physiologically relevant and important step in initiating outer-kinetochore assembly. Another study observed a more modest effect on Dsn1 localization in human cells in which Ser100 and Ser109 had been substituted with Ala, when combined with additional Ser to Ala substitutions at the extreme region of the Dsn1 N terminus, which we cannot model in our structure[32]. We also tested our model of how the Dsn1[N] auto-inhibitory segment interacts with the MIS12C heads. Consistent with our structure and model, the auto-inhibitory peptide of Dsn1 (residues 92–113; Dsn1[92–113]) bound to MIS12C with high-micromolar affinity, an interaction that was importantly enhanced, by 44-fold, on deleting Dsn1[N] from MIS12C (MIS12C[Dsn1ΔN]) (Fig. 2g and Extended Data Fig. 4d,e). The interaction of Dsn1[92–113] with MIS12C was completely abolished by substituting the WRR linchpin of the Dsn1[92–113] peptide with three Ala residues, demonstrating that the WRR motif is essential for engagement of the auto-inhibitory Dsn1[N] segment (Fig. 2g and Extended Data Fig. 4f).

To further assess the role of the Dsn1[N] auto-inhibitory segment in binding between CENP-C and MIS12C in solution, we used analytical size-exclusion chromatography (SEC) (Extended Data Fig. 5a,b). Wild-type MIS12C only partially bound CENP-C residues 1–71 (CENP-C[1–71]), consistent with our structural model in which half of the CENP-C binding interface is blocked. Relief of Dsn1 auto-inhibition, exposing the complete CENP-C-binding site either by deleting Dsn1[N] (MIS12C[Dsn1ΔN]) or introducing substitutions in the WRR linchpin motif of Dsn1[N], resulted in binding of almost all CENP-C, and incorporation of the phosphomimetic MIS12C[Dsn1-S100D S109D] mutants also strongly increased CENP-C binding. These results are in agreement with

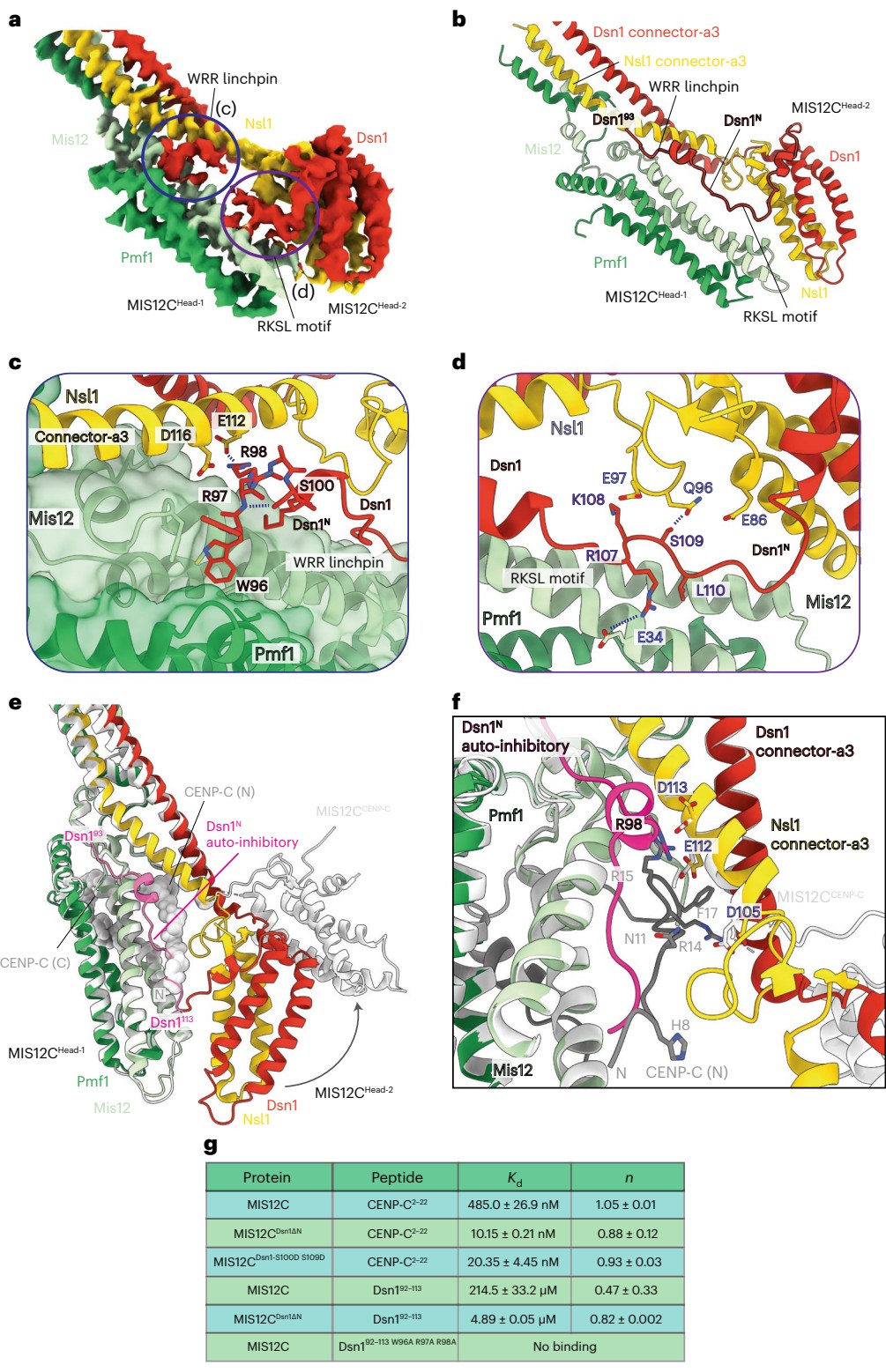

**Fig. 2 | Molecular details of MIS12C auto-inhibition. a**, Cryo-EM density map of MIS12C[Head-1] and MIS12C[Head-2] in the auto-inhibited state; (c) and (d) refer to the views in **c** and **d**, respectively. **b**, Molecular model of the MIS12C head groups in the auto-inhibited state, highlighting the auto-inhibitory Dsn1[N] element. Residues of Dsn1 that are N-terminal to Arg92 are disordered. **c**, Molecular details of the WRR linchpin interaction with the MIS12C[Head-1], with Ser100 forming stabilizing interactions with the backbone of the linchpin loop. **d**, Molecular details of the RKSL zipper binding across MIS12C[Head-1] and MIS12C[Head-2] and locking the auto-inhibited MIS12C state. **e**, The MIS12C–CENP-C complex structure (gray, PDB: 5LSK ref. 10) overlaid over the auto-inhibited MIS12C structure (green, this work). CENP-C is displayed in space-filling representation, and Dsn1[N] is colored red, showing a direct steric clash between CENP-C and Dsn1[N]. The N terminus of CENP-C is indicated with 'N,' and the N-terminal half of CENP-C bound to MIS12C (residues 6–22) is labeled 'CENP-C (N)' and residues 28–48 are labeled 'CENP-C (C).' The boundaries of the Dsn1[N] auto-inhibitory segment (residues 93–113) are labeled and colored pink. **f**, Rotation of MIS12C[Head-2] in transition from the MIS12C auto-inhibited state to the MIS12C–CENP-C complex creates a binding site for CENP-C by removing steric hindrance from both MIS12C[Head-2] and the Dsn1–Nsl1 connector-α3 helices, which also optimizes contacts to Arg14, Arg15 and Phe17 of CENP-C. The N termini of the Dsn1–Nsl1 connector-α3 helices bend slightly on transition to the MIS12C–CENP-C complex. **g**, Summary of the ITC data for dissociation constants ($K_d$) and stoichiometry ($n$) between MIS12C variants and either CENP-C[2–22] or Dsn1[92–113] (Dsn1[N] auto-inhibitory) peptides; experimental data and detailed descriptions are provided in Extended Data Figure 4.

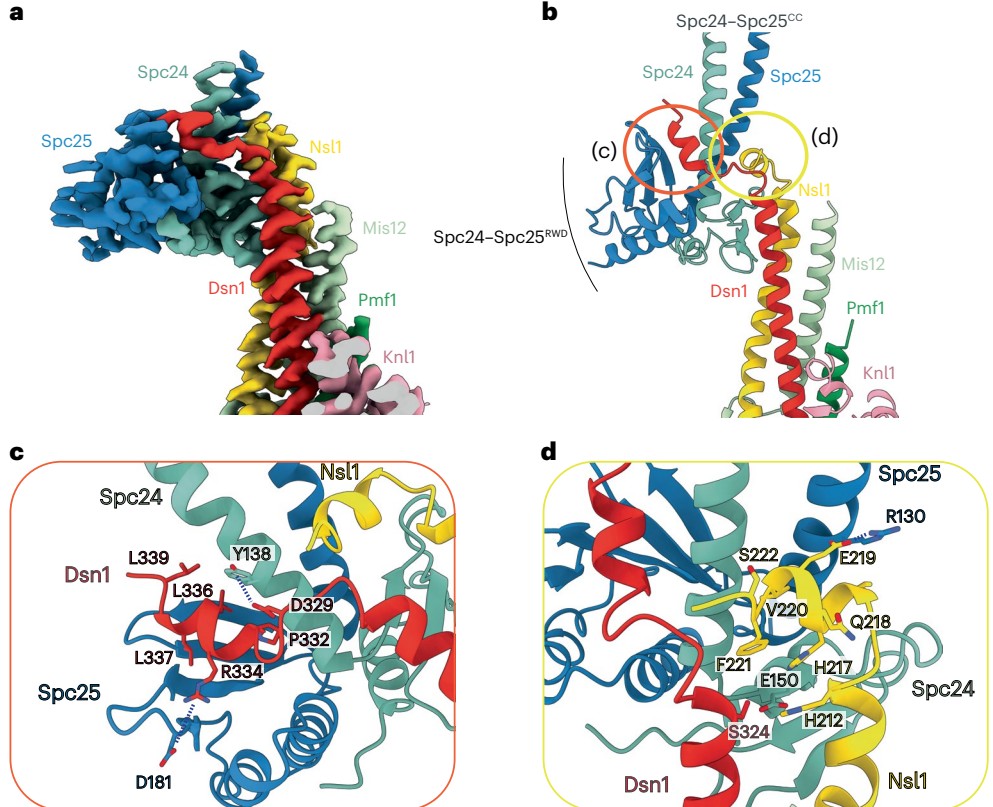

**Fig. 3 | Spc24–Spc25 forms a rigid interface with MIS12C. a**, Cryo-EM density map of the MIS12C–Spc24–Spc25 interface. **b**, Molecular model of the MIS12C–Spc24–Spc25 interface, composed of rigidly docked Spc24–Spc25^RWD formed from both a Dsn1 peptide contact with Spc24–Spc25 and Nsl1 peptide interface with Spc25; (c) and (d) refer to the views in **c** and **d**, respectively. **c**, Molecular details of the Dsn1 peptide interaction with Spc24–Spc25, highlighting a number of electrostatic interactions and the buried hydrophobic surface of Dsn1. **d**, Molecular details of the Nsl1 peptide interaction with Spc24–Spc25, similarly showing an electrostatic and geometric match between the Nsl1 peptide and Spc25 surface.

in vitro MIS12C–CENP-C affinity data reported here and previously[10], localization experiments in cells[35–37] and our structural prediction that phosphorylation of Ser100 and Ser109 relieves Dsn1^N-mediated auto-inhibition to optimize CENP-C binding.

To further understand whether the molecular mechanism of Dsn1 auto-inhibition that we described above is widely conserved, we aligned Dsn1 sequences of several model eukaryotes and observed that both the WRR linchpin and the RKSL motif are highly conserved in vertebrates (Extended Data Fig. 5c). However, these sequence motifs are absent from other distantly related species, including yeast, although the interaction of *Saccharomyces cerevisiae* MIND (the homolog of human MIS12C, termed Mis12^MIND) with CENP-C is also regulated by Aurora B kinase phosphorylation of two equivalent serine residues in Dsn1 (ref. 37). Biochemical studies of *Kluyveromyces lactis* Mis12^MIND support a model in which an unstructured region of Dsn1, including the two Aurora B target sites, auto-inhibits CENP-C binding to Mis12^MIND, and Aurora B kinase phosphorylation of these two sites relieves this auto-inhibition[9]. To understand whether the molecular basis of yeast Mis12^MIND auto-inhibition is evolutionary conserved in human despite sequence divergence, we used AlphaFold2 (ref. 38) to predict the structure of the full-length budding yeast Mis12^MIND complex. Although AlphaFold2 failed to predict the full-length *S. cerevisiae* Mis12^MIND complex, it could predict the related *K. lactis* Mis12^MIND complex structure (*Kl*-Mis12^MIND) (Extended Data Fig. 5d,e). In this prediction, the *Kl*-Dsn1 N terminus binds across the surface of the *Kl*-Mis12^MIND Head-1 domain, bridging it to the Head-2 domain and occluding the hypothetical CENP-C binding interface. The predicted *K. lactis* Mis12^MIND complex auto-inhibition is strikingly similar to that of the structure we observe for human MIS12C, although the amino

acid sequences involved in the interaction have diverged considerably between the two species. Additionally, the structure of the predicted *K. lactis* Dsn1^N auto-inhibitory segment is mainly α-helical, in contrast to the more extended conformation of its human counterpart, precluding a structure-based sequence alignment. Together, our findings suggest that MIS12C auto-inhibition is likely to be an evolutionarily conserved mechanism that regulates inner- and outer-kinetochore assembly in many eukaryotic lineages.

## Dsn1^N likely inhibits the binding of the CENP-T to MIS12C

The molecular details of the interaction between CENP-T, the other major receptor of the KMN network at the centromere, and MIS12C are unknown. This limitation prevents us from unambiguously addressing, on the basis of our structure, how MIS12C auto-inhibition regulates CENP-T binding. Numerous lines of evidence, however, suggest that Dsn1^N also likely inhibits CENP-T–MIS12C association: (1) CENP-T and CENP-C cannot simultaneously bind the same MIS12C protein[17], suggesting that they recognize an overlapping interface, and (2) introducing substitutions into MIS12C that would perturb auto-inhibition, such as deleting MIS12C^Head-2 or altering residues in the vicinity of the Dsn1 RKSL zipper, importantly stimulated CENP-T binding[18]. To further explore this question, we used AlphaFold2 to predict an interaction between CENP-T and MIS12C (Extended Data Fig. 6a). AlphaFold2 predicted that residues 180–235 of CENP-T specifically bind across the MIS12C^Head-1 domain and parts of the MIS12C coiled coils (Extended Data Fig. 6a), consistent with observations that this region is essential for CENP-T-based MIS12C recruitment to kinetochores in cells[14,16,39]. A peptide based on this region also interacts with MIS12C in vitro[18]. Additionally, in the AlphaFold2 model, two key targets of CDK phosphorylation on CENP-T,

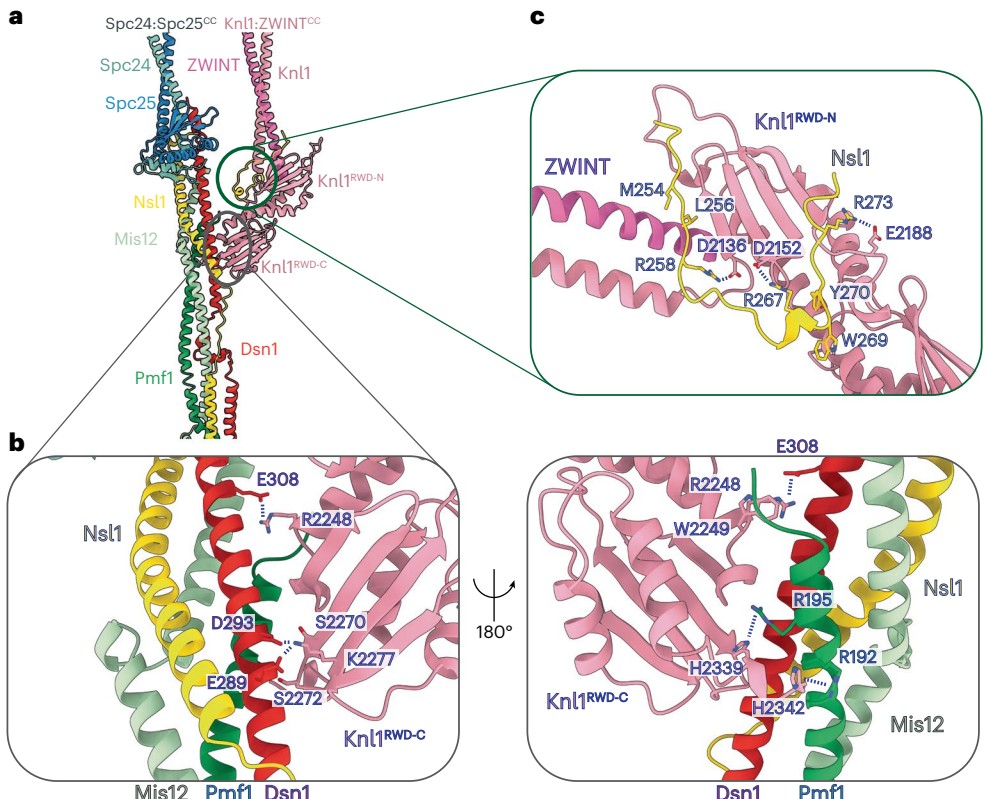

**Fig. 4 | KNL1C forms multiple interactions with MIS12C. a**, Overview of the MIS12C–KNL1C interface. The pair of RWD domains of Knl1 dominate this interface. **b**, Molecular details of the Knl1^RWD-C interaction with MIS12C stalk, formed by Trp2249 docking into shallow grooves of the MIS12C central stalk and

additionally supported by a number of electrostatic interactions on the opposite end of the Knl1^RWD-C. **c**, The Nsl1 C-terminal peptide augments the central Knl1^RWD-C β-sheet interaction with MIS12C, forming additional contacts with Knl1^RWD-N.

Thr195 and Ser201, are in close proximity to basic residues of MIS12C, explaining how their phosphorylation would promote MIS12C–CENP-T interactions[14,16] (Extended Data Fig. 6b). The CENP-T binding site on MIS12C overlaps with Dsn1^N (Extended Data Fig. 6c), demonstrating that MIS12C auto-inhibition indeed is likely to inhibit CENP-T binding, as suggested from prior in vitro data[18]. Notably, Arg209 of CENP-T occupies the same position as Arg98 of Dsn1^N (part of the WRR linchpin) as well as Arg15 of CENP-C, indicating that all three peptide regions recognize a similar small site on MIS12C. Overall, MIS12C auto-inhibition likely regulates CENP-T-based centromere recruitment of MIS12C through a comparable mechanism to that of CENP-C.

### Spc24–Spc25 forms a composite interface with MIS12C
Multiple MIS12C motifs are important for mediating its interaction with NDC80C, particularly the C termini of Dsn1 and Nsl1 (refs. 10,11). A similar Dsn1–NDC80C interaction is conserved in yeast[9,28]. Consistent with these results, we observe an extensive interaction interface between the Spc24–Spc25^RWD domains with the C-terminal coiled coils as well as peptide regions of Dsn1 and Nsl1 (Fig. 3a,b). Specifically, Spc24^RWD rigidly docks onto the top of the Dsn1–Nsl1 coiled coil, which is additionally stabilized by the Mis12 α-helix on the back side (Fig. 3b). The C-terminal region of Dsn1 folds across Spc25^RWD (Fig. 3c), and Nsl1 forms an α-helix abutting the Spc24–Spc25 coiled coils (Fig. 3b,d). Overall, Dsn1–Nsl1 forms an extended and intertwined connection with Spc24–Spc25, secured by multiple points of contact that might be necessary to withstand forces of the mitotic spindle across the MIS12C–NDC80C interface. Consistent with our structural model, altering either the Dsn1 contact site (Dsn1-P332W R334A L336R mutant) (Fig. 3c) or the Nsl1 contact site (Nsl1-E219R V220R F221A mutant) (Fig. 3d) reduced the affinity of MIS12C for NDC80C, as assessed by SEC

experiments (Extended Data Fig. 7a,b). Combining the Dsn1-P332W R334A L336R and Nsl1-E219R V220R F221A mutants to disrupt both interfaces completely abolished MIS12C–NDC80C interactions, consistent with two independent points of contact between MIS12C and NDC80C (Extended Data Fig. 7a,b). In the accompanying manuscript, Polley et al.[40] show that introducing substitutions into either of the the Dsn1 or Nsl1 interfaces that contact Spc24–Spc25 importantly reduced NDC80C localization to kinetochores.

Comparison of the available crystal structures of the CENP-T–Spc24–Spc25^RWD complex with our MIS12C–Spc24–Spc25^RWD reconstruction shows that CENP-T binds Spc24–Spc25^RWD at exactly the same interface as the Dsn1 subunit of MIS12C (Extended Data Fig. 7c)[28,29]. These results are in agreement with a prior crystal structure of yeast Spc24–Spc25 in complex with a peptide modeled on the Spc24–Spc25-binding site of Dsn1, which showed Dsn1 and CENP-T share the same binding site on Spc24–Spc25 (ref. 10). This is consistent with CENP-T and MIS12C functioning in separate pathways to recruit NDC80C independently of one another. Interestingly, chicken CENP-T effectively combines both Dsn1 and Nsl1 C-terminal regions into a single polypeptide to bind across the entire Spc24–Spc25^RWD (Extended Data Fig. 7c).

### Knl1^RWD rigidly docks onto the MIS12C
The ordered C-terminal region of Knl1 consists of a tandem pair of RWD domains that transition into the Knl1–ZWINT coiled coil (Fig. 1b,e and Fig. 4a). Knl1^RWD-C rigidly docks and recognizes a shallow surface formed by the Dsn1 and Pmf1 coiled coils, whereas Knl1^RWD-N uses a canonical RWD protein-interaction groove to bind the otherwise disordered Nsl1 C-terminal tail (Fig. 4a–c and Extended Data Fig. 8a). Overall, in the context of the KMN complex, each RWD domain of Knl1 engages a

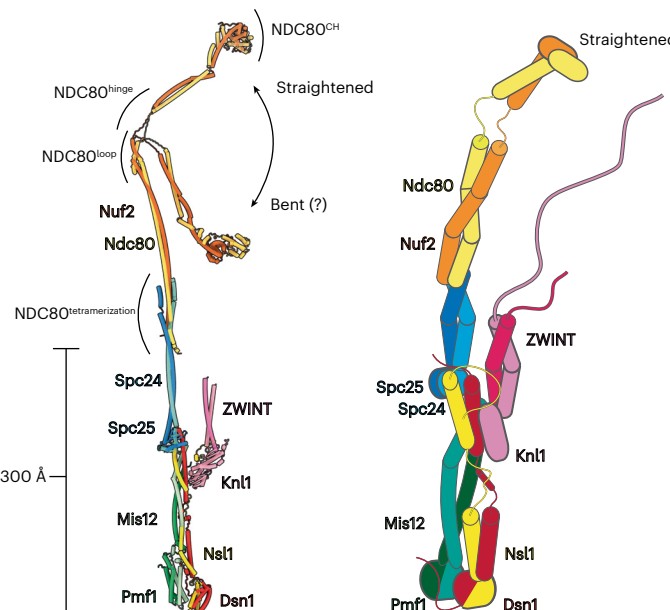

**Fig. 5 | Molecular model of the complete human KMN network complex.** Molecular model and cartoon schematic of the complete KMN network complex based on the structures determined in this study and AlphaFold2 models. The existence of the bent conformation of NDC80C is not firmly established. NDC80CH, NDC80 calponin homology domain; NDC80tetramerization, NDC80C tetramerization domain.

distinct part of MIS12C, explaining why Knl1 in most species contains two connected RWD domains. Mechanistically, this would provide robustness to the Knl1–MIS12C interaction: if one of the connections fails, the Knl1–MIS12C complex will likely remain intact because of the second redundant, independent linkage.

Knl1RWD-C complements the surface and charge of the MIS12C tetrameric coiled coil (Fig. 4a,b). Specifically, Arg2248 and Trp2249 of Knl1 insert into a pocket formed by the Dsn1 and Pmf1 subunits, an interaction that is additionally stabilized by Dsn1E308 and the C-terminal tail of Pmf1 (Fig. 4b). On the opposite end, Knl1RWD-C uses a series of electrostatic locks to engage the complementary surface charges of MIS12C (Fig. 4b).

Our use of full-length proteins allowed us to observe a long peptide of the Nsl1 C terminus bound to Knl1RWD-N (Fig. 4c). In addition to the previously reported nine-residue Nsl1266–274 peptide bound to Knl1RWD-N (ref. 11), we observe that almost the entire C terminus of Nsl1250–277 (28 residues) binds across the extended interface formed by Knl1RWD-N and the Knl1–ZWINT coiled coils, with part of the interactions provided directly by ZWINT. A large portion of this interaction is well resolved in our reconstruction, with the side chains of amino acids being clearly visible (Extended Data Fig. 8a). A portion of this interface is seen only at lower resolution in unsharpened cryo-EM maps (Extended Data Fig. 8b). To further confirm our assignment and modeling, we used AlphaFold2 to predict the Knl1–ZWINT–Nsl1C terminus complex (Extended Data Fig. 8c). In five out of five models, the AlphaFold2 model matched our experimental structure with high confidence, including Knl1–ZWINT coiled coil modeling, further validating our assignment. Overall, Nsl1 uses two highly conserved motifs at its C terminus to bind to the Knl1–ZWINT complex: a YPLR motif binds to Knl1RWD-N, and a VLKRK motif binds across the Knl1–ZWINT coiled coil (Extended Data Fig. 8a,b,d). The Nsl1 C terminus buttresses the Knl1–ZWINT coiled coils, likely stabilizing the rigid projection of these coiled coils away from the junction.

Deletion of the Nsl1 C terminus (MIS12CNsl1ΔC: residues 265–281 in Nsl1 protein were deleted), which engages Knl1RWD-N, reduced the interaction strength between Knl1 and MIS12C, as assessed by SEC assays. By contrast, modifications to the Knl1RWD-C surface (Knl1mut1: R2248D

W2249A and Knl1mut2: S2270R S2272W) did not importantly disrupt MIS12C–Knl1 interactions (Extended Data Fig. 9). There was little to no change in the interaction between MIS12C and the Knl1 mutant in which the entire Knl1RWD-C surface that interacts with MIS12C was altered (Knl1mut3: S2270R S2272W R2248A W2249A) (Extended Data Fig. 9). This observation agrees with prior data showing that the affinity of Knl1 for the whole MIS12 complex is only slightly higher than its affinity for a peptide modeled on the C terminus of Nsl1 alone (residues 258–281), indicating that Knl1RWD-N dominates Knl1–MIS12C interactions[11]. We found that interfering with both interfaces between MIS12C and Knl1 simultaneously completely abolished the interaction between MIS12C and Knl1 (Extended Data Fig. 9), consistent with our structural model revealing two points of contact between MIS12C and KNL1C.

Previous work showed that disrupting the Knl1RWD-N interaction with the Nsl1 C terminus in cells abolished localization of Knl1 to kinetochores[11]. The accompanying study[40] shows that mutating the Knl1RWD-C surface to prevent Knl1 interactions with the MIS12C stalk also prevents Knl1 localization to kinetochores. Together, these results indicate that both Knl1RWD-N and Knl1RWD-C are necessary for Knl1 recruitment to kinetochores in cells.

### Overall model of the KMN network

In our cryo-EM reconstructions, we could observe only the well-resolved and conformationally rigid KMNjunction module, including the C terminal RWD domains and coiled coil of Spc24–Spc25 (Fig. 1a,b,d). The Spc24–Spc25–Ndc80–Nuf2 tetramerization junction and the Ndc80–Nuf2 subunits were not visible in cryo-EM density (Fig. 1d,e). This could be due to either conformational heterogeneity or denaturation of these regions on the cryo-EM grid. Moderate flexing of the Spc24–Spc25 coiled coil combined with minor conformational variability at the Spc25–Spc25RWD–MIS12C interface could account for the gradual reduction of cryo-EM density signal towards the Spc24–Spc25–Ndc80–Nuf2 junction, some 150 Å from the Spc24–Spc25RWD–MIS12C interface. Conformational heterogeneity would be consistent with rotary shadowing EM of NDC80C, which showed both long, straight particles as well as bent particles[17], and also with the measurement of the Ndc80–Nuf2 bending in cells and in solution[33,41]. To model a complete KMN network complex, we used previously published structures[21,23,24], an AlphaFold2 prediction of Ndc80–Nuf2 and the high-resolution structure of the KMNjunction determined in this work (Fig. 5). AlphaFold2 predicted a bent state for the Ndc80–Nuf2 complex, with bending being mediated by the hinge region. We could extrapolate this model to the fully straightened state, allowing us to determine two complete models of the human KMN network complex, with the majority of details known at high resolution (Fig. 5). Our model is remarkably consistent with outer-kinetochore dimensions measured in cells[33,42]. The NDC80C bent conformation matches the KMN dimensions measured in cells in the absence of microtubule tension, whereas the extended NDC80C conformation aligns well with the microtubule-bound KMN state. However, we do not have sufficient experimental evidence to confidently state that the bent state exists in solution. In our model, KMNjunction represents a rigid, 300-Å-long central scaffold upon which more flexibly tethered components assemble. We speculate that having a central rigid scaffold with defined dimensions is important to introduce a set distance between the chromatin-proximal MIS12C and microtubule-proximal NDC80C, to ensure precise positioning of the error correction and SAC components with respect to their regulation targets. For example, phosphorylation of targets of the major error-correction kinase Aurora B, located at the inner kinetochore, is exquisitely dependent on the distance from the kinase[43]. KMNjunction would provide a defined separation of Aurora B targets away from the kinase, enabling a robust error-correction function by spatial separation.

### Discussion

We determined the cryo-EM structure of the central component of the human outer kinetochore, which revealed a rigid prong-shaped

arrangement of eight subunits of the ten-subunit KMN network complex. Our results and conclusions on the architecture of the KMN network agree with the co-submitted study by Musacchio and colleagues[40]. Their study also showed that the localization of Knl1 and NDC80C to kinetochores in cells was disrupted by structure-based substitutions at their respective interfaces with MIS12C. At the core of the KMN structure is MIS12C, which interacts with the microtubule-binding NDC80C and KNL1C at the tip of its tetrameric coiled coil. These interactions rigidly project coiled coils of Spc24–Spc25 and Knl1–ZWINT, which are perfectly parallel to one another, away from the central MIS12C stalk and inner-kinetochore recruitment sites. This arrangement precisely defines the centromere-microtubule axis along the KMN network and generates defined distances and polarity between the inner- and outer-kinetochore components. Our structure is consistent with such distances measured between different kinetochore components in cells[33,42]. Additionally, we observe numerous independent interactions between NDC80C and MIS12C, as well as between KNL1C and MIS12C, that are likely to have redundant functions and cooperate to provide strength and robustness to the KMN network as a central load-bearing component of the outer kinetochore, explaining how it can withstand mitotic forces without detachment. We validated all new interfaces with specific point substitutions using biochemical and biophysical assays, allowing us to decouple the contribution of each interface to KMN complex assembly. Our structural work provides the foundation for cell-biology experiments to understand the contribution of each KMN component to kinetochore and centromere function in cells.

At the centromere-proximal end of the MIS12C, we observed an auto-inhibited MIS12C^Head domain with inaccessible inner-kinetochore binding sites. The auto-inhibited state is directly stabilized by Ser100 and Ser109, two key residues phosphorylated by centromeric Aurora B[16,32,34–37]. Phosphorylation of both of these residues disrupts the MIS12C^Head auto-inhibited state and allows the KMN network to bind the inner kinetochore with high affinity. The initial weak binding between CENP-C and MIS12C observed in our study and by others[10,13] likely allows KMN to be recruited to the inner kinetochore, at which this attachment is stabilized by Aurora B-based phosphorylation. We speculate that MIS12C auto-inhibition allows outer-kinetochore assembly specifically at the functional centromeres, and we propose that this mechanism prevents formation of active kinetochores at non-centromeric loci in cells, increasing the fidelity of the chromosome-segregation process. Aurora B activity is also necessary for the error-correction pathway that ensures accurate chromosome bi-orientation by resetting KMN-microtubule attachments. KMN network auto-inhibition likely also ensures that complete kinetochore assembly occurs only at sites with active error-correction machinery.

We propose a structural model for the interaction of CENP-T with MIS12C that explains how the interaction is regulated by CDK1-phosphoryation of CENP-T[14,16], how CENP-T–MIS12C binding and CENP-C–MIS12C binding are mutually exclusive[17] and how Aurora B kinase phosphorylation of Dsn1^N relieves auto-inhibition to importantly stimulate CENP-T binding[18]. Therefore, MIS12C auto-inhibition likely evolved to regulate both CENP-C- and CENP-T-based centromere-recruitment pathways.

## Online content

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

## Methods

### Cloning of the KMN network complexes

Genes encoding *Homo sapiens* Ndc80, Nuf2, Spc24, Spc25, Mis12, Pmf1, Dsn1 and Nsl1 were synthesized by Thermo Fisher Scientific, with codon optimization for *Trichoplusia ni*. The coding fragments of Knl1 and ZWINT were amplified by PCR from cDNA and cloned into a pU1 plasmid[44]. NDC80C, MIS12C and KNL1C were subsequently cloned separately into a modified Multibac expression system[44]. TEV cleavable double strep II (DS) tags were added to the C termini of Nuf2 and Pmf1. KNL1C was subsequently re-cloned to generate the Knl1 1870–2342 fragment with an N-terminal 6×His-SNAP tag. The constructs for Dsn1$^{\Delta N}$ (6–113 residues deleted), Dsn1$^{P332W R334A L336R}$, Nsl1$^{1–264}$, Nsl1$^{E219R V220R F221A}$ and Dsn1$^{W96A R97A R98A}$ were cloned into the plasmid containing the Nsl1–Dsn1 pair, and the proteins were expressed in combination with the second virus encoding Mis12–Pmf1 proteins.

Synthetic genes for *H. sapiens* Knl1$^{2131–2337}$ and CENP-C$^{1–71}$, fused to a C-terminal maltose-binding protein (MBP) tag (separated by a 5′-AACGCCGCCAGCGGT-3′ linker), were supplied by Integrated DNA Technologies. Both genes were inserted into pET47b plasmids carrying a kanamycin selection marker, which expresses a 3C cleavable N-terminal 6×His tag. Mutants of Knl1$^{2131–2337}$, namely the two double mutants R2248D W2249S and S2270R S2272W and the quadruple S2270R S2272W R2248A W2249A mutant, were created with the QuikChange protocol, using primers supplied by Merck.

### Expression and purification of the KMN network and CENP-C components

The baculoviruses for expression of all KMN network complexes were generated using standard protocols[44]. All the KMN network complexes were expressed individually in High-5 insect cells (BTI-TN-5B1-4, Thermo Fisher Scientific). The High-5 insect cell line was not tested for mycoplasma contamination and was not authenticated. Typically, 4 l of High-5 cells were infected 2.5% vol/vol with the P3 cell culture, and the cells were collected by centrifugation 48–72 h after infection.

The cells expressing NDC80C, MIS12C or their mutants were lysed with a sonicator in lysis buffer (50 mM Tris-HCl pH 8.0, 300 mM NaCl, 0.5 mM TCEP and 5% glycerol) supplemented with benzamidine, EDTA-free protease inhibitor tablets and benzonase. Clarified lysate was loaded onto a Strep-Tactin column (Qiagen), immobilized proteins were washed with lysis buffer and the complexes were eluted in a buffer containing 50 mM Tris-HCl pH 8.0, 300 mM NaCl, 1 mM TCEP, 5% glycerol and 5 mM desthiobiotin (Sigma).

For the NDC80C, the Strep-Tactin eluate was diluted to a final salt concentration of 75 mM NaCl and loaded onto the Resource Q anion exchange column (Cytiva). The protein was eluted in 20 mM HEPES pH 8.0, 1 mM TCEP and 5% glycerol, with a NaCl gradient from 75 mM to 600 mM. The peak fraction was concentrated and further purified using a Superdex 200 16/600 column (Cytiva) in gel filtration buffer (20 mM HEPES pH 7.8, 150 mM NaCl, 1 mM TCEP). The peak NDC80C fractions were concentrated and flash frozen in liquid nitrogen.

For MIS12C, the Strep-Tactin eluate was diluted with 50 mM Tris-HCl pH 7.4 to a final salt concentration of 50 mM NaCl and directly loaded onto a Resource S anion exchange column (Cytiva). The protein was eluted in 20 mM Tris-HCl pH 7.4, 1 mM MgCl$_2$, 1 mM TCEP and 5% glycerol, with an NaCl gradient from 50 mM to 500 mM. The peak fraction was concentrated and further purified using a Superdex 200 16/600 column (Cytiva) in MIS12C gel filtration buffer (20 mM HEPES pH 7.8, 150 mM NaCl, 1 mM MgCl$_2$, 1 mM TCEP). The peak MIS12C fractions were concentrated and flash frozen in liquid nitrogen.

Insect cells expressing KNL1C were lysed with a sonicator in KNL1C lysis buffer (50 mM Tris-HCl pH 8.0, 300 mM NaCl, 0.5 mM TCEP, 10 mM imidazole, 1 mM EDTA and 10% glycerol) supplemented with benzamidine, EDTA-free protease inhibitor tablets and benzonase. Clarified lysate was loaded onto a cOmplete His-tag purification column (Roche), immobilized proteins were washed with lysis buffer and the complexes were batch eluted in a buffer containing 50 mM Tris-HCl pH 8.0, 300 mM NaCl, 0.5 mM TCEP, 10 mM imidazole, 1 mM EDTA and 10% glycerol supplemented with 300 mM imidazole (Sigma).

NP-40 was added to the Knl1–ZWINT eluate to a final concentration of 0.01%, and the eluate was diluted to a final salt concentration of 150 mM NaCl and loaded immediately onto a Resource Q anion exchange column (Cytiva). The protein was eluted in 20 mM HEPES pH 8.0, 1 mM TCEP and 10% glycerol, with a NaCl gradient from 150 mM to 600 mM. The eluted complex was immediately flash frozen in liquid nitrogen.

CENP-C$^{1–71}$ with an MBP tag at its C terminus and Knl1$^{2131–2337}$ and its mutants were expressed in *E. coli* and purified. Plasmids were transformed into *E. coli* BL21 (DE3) cells (Thermo Fisher Scientific) and three 1-l cultures were induced at an optical density at 600 nm of 0.6 with 0.3 mM IPTG and grown for 18 h at 18 °C. Overexpressed proteins were purified by Ni-NTA(II) affinity chromatography (HisTrap HP, Cytiva), followed by the reverse-IMAC strategy after cleavage of the N-terminal 6×His tag using a recombinant 3C protease carrying a 6×His tag. A final purification step was performed using a Hi-load Superdex 75 gel filtration column (Cytiva) equilibrated with 10 mM Tris-HCl pH 8.0 and 150 mM NaCl. The eluted proteins were immediately flash frozen in liquid nitrogen.

### Reconstitution of the KMN network complex for cryo-EM

NDC80C, MIS12C and KNL1C were rapidly thawed, mixed in equimolar amounts (1:1:1 ratio) and applied to a Superose 6 3.2/300 (Cytiva) gel filtration column in buffer containing 20 mM HEPES pH 8.0, 80 mM NaCl, 1 mM MgCl$_2$ and 1 mM TCEP. The peak fractions were analyzed by SDS–PAGE and concentrated to 1 mg ml$^{-1}$.

### Cryo-EM grid preparation and data acquisition

Three microliters of prepared complexes at a concentration of approximately 1 mg ml$^{-1}$ were applied to UltrAuFoil 300 mesh gold R1.2/1.3 grids (Quantifoil Micro Tools) that had been freshly glow discharged using an Edwards S150B glow discharger for 75 s at setting 6, 30–35 mA, 1.2 kV and 0.2 mBar (0.15 Torr). The grids were then flash frozen in liquid ethane using a Thermo Fisher Scientific Vitrobot IV (0–0.5 s waiting time, 2 s blotting time, −7 blotting force) using Whatman filter paper 1. Cryo-EM images were collected using a Thermo Fisher Scientific Titan Krios microscope operating at 300 keV using a Gatan K3 camera. A magnification of ×81,000 was used, yielding pixel sizes of 1.059 Å px$^{-1}$. The images were recorded at a dose rate of 16 e$^-$ px$^{-1}$ s$^{-1}$, with an exposure time of 3.2 s and 45 frames. The Thermo Fisher Scientific automated data-collection program EPU was used for data collection using aberration-free image shifts (AFIS). Defocus values ranged from −1.2 to −2.4 μm, at an interval of 0.2 μm.

### Cryo-EM data processing

Micrograph video frames were aligned with MotionCor2 (ref. 45), and contrast-transfer function (CTF) estimation was performed by CTFFIND[46], as integrated into RELION 4.0 (ref. 47). Particle picking was performed using TOPAZ[48]. Default parameters were used for data processing, unless stated otherwise.

Particles were initially subjected to 2D classification in cryoSPARC[49] into 100 classes using 30 online EM iterations with 400 batch sizes per class. Ab initio models were further refined individually to obtain better-defined cryo-EM densities using homogeneous refinement and standard settings. Heterogeneous refinement in cryoSPARC with the refined KMN network volume and five decoy noise volumes was performed against the entire set of automatically picked particles using default parameters. Particles corresponding to the KMN network complex were further refined using a non-uniform refinement job type. A clean particle stack was then exported back into RELION 4.0, and two successive rounds of CTF refinement and Bayesian polishing were performed. A final, polished particle data set was used for a final round of 3D refinement in RELION, yielding a final resolution of 3.0 Å (GS-FSC).

To generate cryo-EM densities with a better-resolved MIS12C head group, a mask around the MIS12C head groups was generated, and classification without alignment was performed in RELION 4.0 (Extended Data Fig. 2). The class with high MIS12C$^{Head-2}$ occupancy was selected and further refined, giving a reconstruction at a 3.4-Å resolution (GS-FSC). This reconstruction was further subjected to multi-body refinement using a previously generated mask around the MIS12C head groups. This resulted in a well-resolved MIS12C head groups at a resolution of 3.8 Å (GS-FSC).

### Cryo-EM model building and refinement

AlphaFold2 models for NDC80C and KNL1C, generated using full-length sequences, were docked into the cryo-EM densities by rigid body fitting in ChimeraX[50] and manually corrected and adjusted in COOT[51], including rebuilding regions ab initio. The models were real-space refined in PHENIX[52] against consensus and composite maps. The Spc24–Spc25–MIS12C–Knl1 core was refined against the consensus map determined at 3.0-Å resolution, whereas the MIS12C$^{Head-1}$–MIS12C$^{Head-2}$ was refined against Body 2, determined at 3.8-Å resolution (Extended Data Fig. 2c). The composite map of the KMN$^{Junction}$ was reconstructed in RELION 4.0 using MultiBody refinement, where Body 1 and Body 2 were combined in the same coordinates. MultiBody refinement was used to improve local resolution around Spc24–Spc25 coiled coils, as well as the Knl1–ZWINT coiled coils (Extended Data Fig. 3a). The α-helical sequence register was assigned by examining the higher resolution at the base of the coiled coils, where amino acid side chains are apparent. Helices were traced until the point at which cryo-EM density disappeared. Model validation was performed against the consensus map determined at 3.0-Å resolution. Default parameters for structure refinements in PHENIX were used, including secondary structure and Ramachandran constraints. Figures were prepared using ChimeraX[50].

### AlphaFold2 predictions

AlphaFold2[38] predictions were run using locally installed versions of the program. All the sequences used are canonical proteins sequences obtained from Uniprot.

### ITC experiments

Peptides were synthesized by Cambridge Research Biochemicals (Dsn1$^{92–113}$) and Biosynth (CENP-C$^{2–22}$), with N-terminal acetylation and C-terminal amidation. Sequences were as follows: wild-type Dsn1$^{92–113}$, RRQS*WRR*ASMKETNRRKSLHPI-AW; mutant Dsn1$^{92–113}$, RRQS*AAA*ASMKETNRRKSLHPI-AW (substituted residues are italicized; Ala-Trp were added for determination of concentration by absorption at 280 nm); CENP-C$^{2–22}$, AASGLDHLKNGYRRRFCRPSRA. ITC was performed using an Auto-iTC200 instrument (Malvern Instruments) at 20 °C. Before the ITC runs, all components were dialyzed for at least 16 h against ITC buffer: 20 mM HEPES (pH 7.5), 100 mM NaCl and 1 mM TCEP at 4 °C. For each titration run, 370 µl of protein sample was loaded in the calorimeter cell. Either the CENP-C$^{2–22}$ peptide or Dsn1$^{92–113}$ peptide (wild type or mutant) was titrated into the cell in one 0.5-µl injection, followed by 19 injections of 2 µl each. The cell contained MIS12C variants (wild type, MIS12C$^{Dsn1-S100D S109D}$, MIS12C$^{Dsn1ΔN}$). After the initial injection was discarded, the changes in the amount of heat released were integrated over the entire titration and fitted to a single-site binding model using the MicroCal PEAQ-ITC Analysis Software 1.0.0.1258 (Malvern Instruments). Details of the concentration of components in the syringe and cell are provided in Extended Data Figure 4. Titrations were performed in either duplicate or triplicate.

### Assessment of substitutions in the MIS12C–NDC80C interfaces

To test the effect of substitutions in the MIS12C–NDC80C interfaces, either wild-type MIS12C or three MIS12C mutants (MIS12C$^{Nsl1-E219R V220R F221A}$, MIS12C$^{Dsn1-P332W R334A L336R}$, MIS12C$^{Nsl1-E219R V220R F221A, Dsn1-P332W R334A L336R}$) were

mixed with wild-type NDC80 at 4 µM each in a buffer of 20 mM HEPES (pH 7.8), 150 mM NaCl and 0.5 mM TCEP, and 50 µl was loaded onto a Superose 6 size-exclusion column (Cytiva).

### Assessment of substitutions in the MIS12C–KNL1 interfaces

To test the effect of substitutions in the MIS12C–KNL1 interfaces, either wild-type MIS12C or MIS12C$^{Nsl1ΔC}$ was mixed with either wild-type KNL1 or two mutants (KNL1$^{R2248D W2249A}$ (KNL1$^{mut1}$), KNL1$^{S2270R S2272W}$ (KNL1$^{mut2}$)) at 4 µM each in a buffer of 20 mM HEPES (pH 7.8), 150 mM NaCl and 0.5 mM TCEP, and 50 µl was loaded onto a Superose 6 size-exclusion column (Cytiva).

### Testing the effects of the Dsn1 auto-inhibitory segment on CENP-C$^{1–71}$ binding to MIS12C

Four micromolar of CENP-C$^{1–71}$-MBP was mixed with 4 µM of either wild-type MIS12C or three mutants (MIS12C$^{ΔN-Dsn1}$, MIS12$^{Dsn1-W96A R97A R98A}$, MIS12C$^{Dsn1-S100D S109D}$) in a buffer of 20 mM HEPES (pH 7.8), 150 mM NaCl and 0.5 mM TCEP, and 50 µl of the mixture was loaded onto a Superdex S200 size-exclusion column (Cytiva).

### Reporting summary

Further information on research design is available in the Nature Portfolio Reporting Summary linked to this article.

## Data availability

The cryo-EM maps generated in this study have been deposited in the Electron Microscopy Data Bank (EMDB) under the accession code EMD-17814. The atomic coordinates have been deposited in the PDB under the accession code 8PPR. Publicly available PDB entries used in this study are: 5LSK, 3VZA, and AlphaFold2 AF-Q8NG31-F1. CENP-C uniprot entries used for sequence alignments: Q03188 (*H. sapiens*), P49452 (*M. musculus*), O57392 (*G. gallus*), Q66LI3 (*B. taurus*); DSN1 uniprot entries used for sequence alignment: Q9H410 (*H. sapiens*), A6QQ81 (*B. taurus*), Q9CYC5 (*M. musculus*), Q1T768 (*G. gallus*), A0A8J0US97 (*X. laevis*). Source data are provided with this paper.

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

## Acknowledgements

We are grateful to the LMB EM Facility for help with the EM data collection, J. Grimmett, T. Darling and I. Clayson for computing and J. Shi for help with insect cell expression. We would also like to thank members of Barford group for their input and useful discussions, especially K. W. Muir and N. Turner for critical reading of the manuscript. We also thank A. Musacchio for sharing unpublished experimental observations. This work was supported by UKRI/ Medical Research Council MC_UP_1201/6 (D. Barford), Cancer Research UK C576/A14109 (D. Barford) and Boehringer Ingleheim Fonds Fellowship (S.Y.). The funders had no role in study design, data collection and analysis, decision to publish or preparation of the manuscript. For the purpose of open access, the authors have applied a CC BY public copyright license to any author accepted manuscript version arising.

## Author contributions

D. Barford and S.Y. designed the study and experiments. Z. Z., D. Bellini and S. Y. cloned constructs. J. Y., D. Bellini and S. Y. purified all proteins. S. Y. prepared the grids, collected and processed the cryo-EM data. J. Y. performed all biochemical experiments. D. Barford performed ITC experiments. D. Barford and S. Y. wrote the manuscript, with input from all authors.

## Competing interests

The authors declare no competing interests.

## Additional information

**Extended data** is available for this paper at https://doi.org/10.1038/s41594-024-01249-y.

**Correspondence and requests for materials** should be addressed to Stanislau Yatskevich or David Barford.

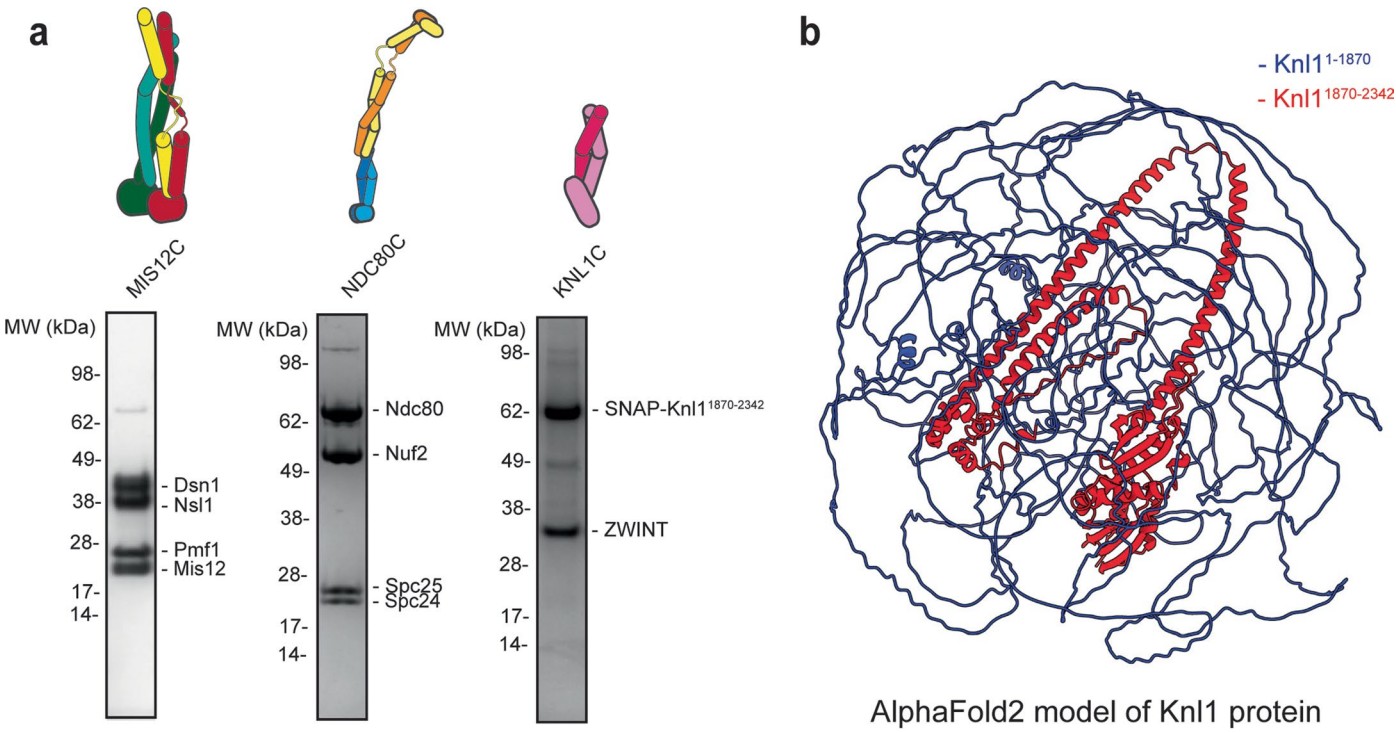

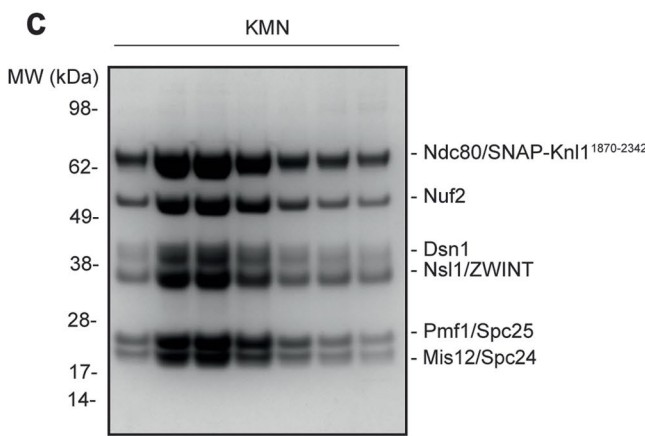

**Extended Data Fig. 1 | Cryo-EM sample preparation. a**. Coomassie blue–stained SDS-PAGE gels of the purified KMN network components and their corresponding schematics that were used in this study for structure-function analysis. The purifications were repeated at least three independent times with identical results. **b** AlphaFold2 prediction of the full-length Knl1 protein obtained from Uniprot server. The Knl1 region coloured red (amino acids 1870–2342) was used in this study as it contains all structured regions of the Knl1 protein. **c** Coomassie blue-stained SDS-PAGE gel of the reconstituted KMN network complex that was used for cryo-EM grid preparations and structural analysis. The reconstitutions were repeated at least five independent times with identical results.

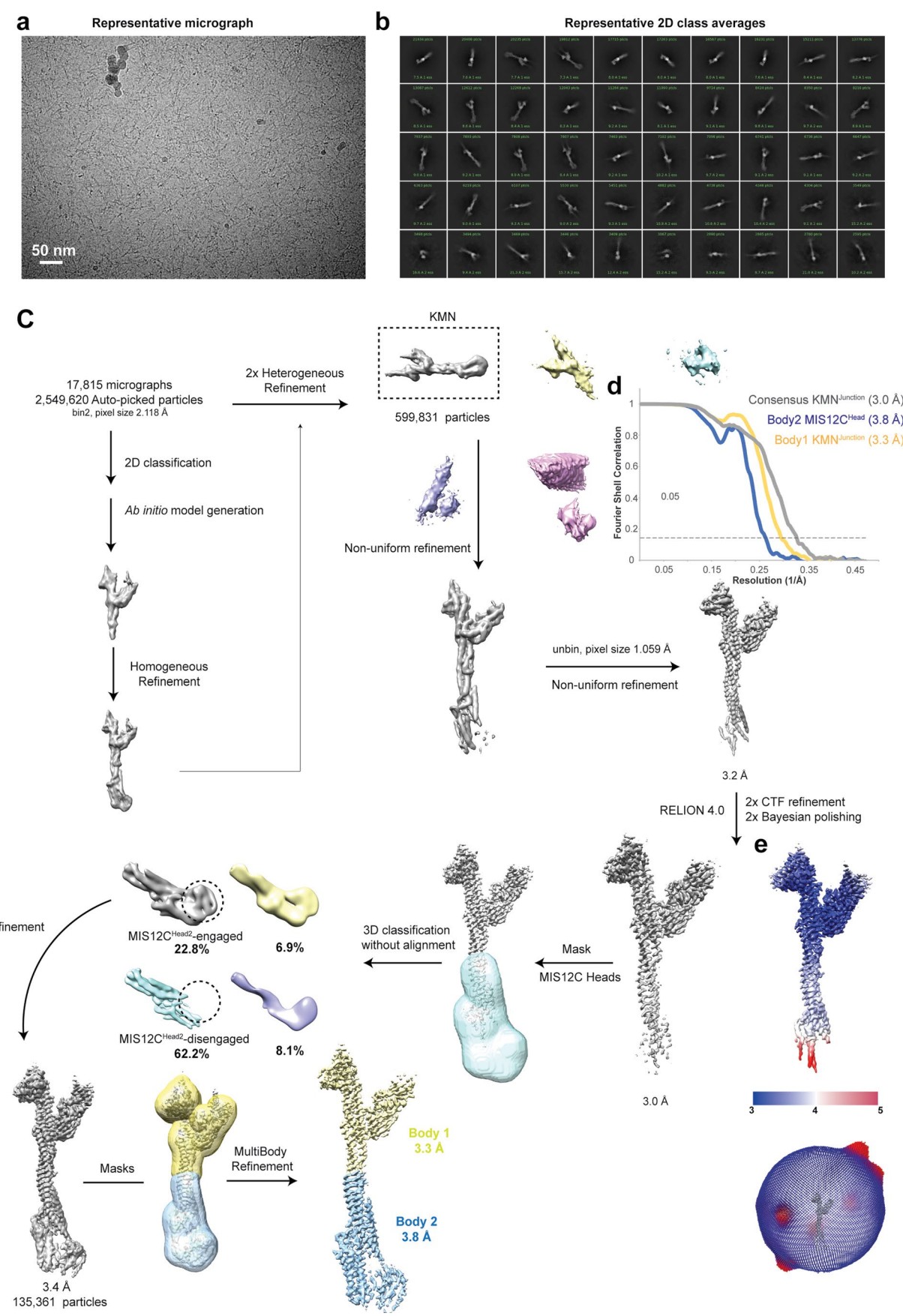

**Extended Data Fig. 2 | See next page for caption.**

**Extended Data Fig. 2 | Cryo-EM data processing. a** Representative cryo-electron micrograph obtained during data collection. A total of 17,815 micrographs were obtained during data collection. **b** Representative 2D class averages generated in cryoSPARC during initial 2D classification steps. **c** Cryo-EM data processing workflow summary as described in the Methods section.

**d** Fourier Shell Correlation (FSC) plots of the three main maps used to generate molecular models. The maps are also highlighted in (c). **e** Consensus KMN$^{\text{Junction}}$ complex coloured by local resolution as well as angular distribution of particle views.

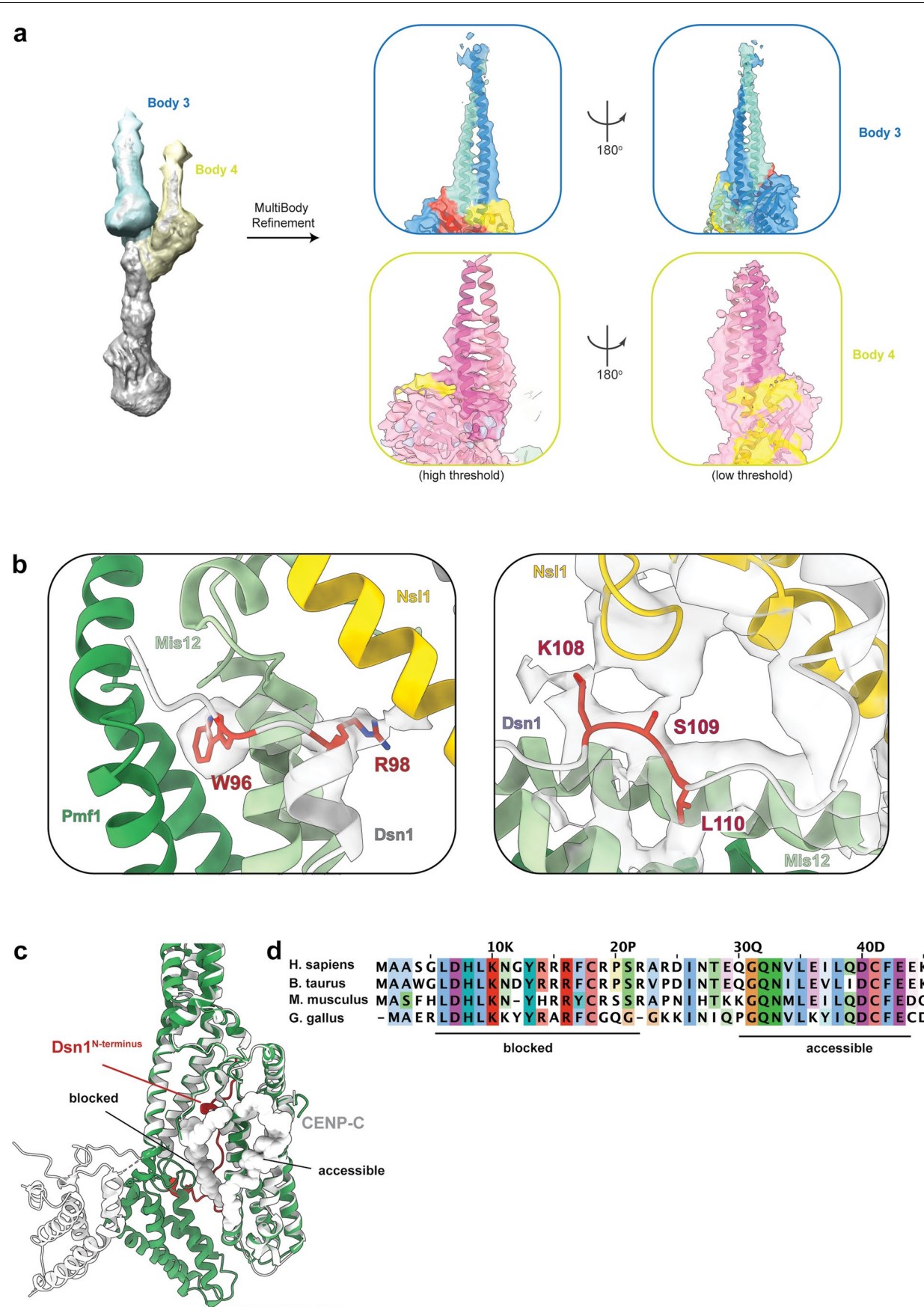

**Extended Data Fig. 3 | See next page for caption.**

**Extended Data Fig. 3 | Molecular details of Dsn1 auto-inhibition. a** Unmasked map of the KMN network complex was used to generate masks around Spc24:Spc25 coiled-coils (Body 3) and independently for Knl1:ZWINT coiled-coils (Body 4). The masks and maps were subsequently used for MultiBody refinement as implemented in RELION 4.0. MultiBody refinement improved local resolution to enable tracing of the coiled-coils for both bodies, as shown in the insets on the right where molecular model of the coiled-coils was fitted into a transparent cryo-EM density map of the Body 3 and Body 4. **b** Molecular model of the Dsn1$^N$ fitted into the MIS12C$^{Head-2}$ cryo-EM density map, showing side-chain resolution at the key interaction regions: Left: view showing side chain density for Trp96 and Arg98. Right: a rotated view showing density for Lys108, Ser109 and Leu110. **c** MIS12C:CENP-C structure (grey, PDB ID: 5LSK ref. 10) overlayed with the auto-inhibited MIS12C shows that the back side of the MIS12C is available to bind a portion of the CENP-C (residues 1–21). **d** CENP-C multiple sequence alignment[53], highlighting the CENP-C regions that can bind to the MIS12C in the auto-inhibited state (accessible region, C-terminal portion) and the region of CENP-C (N-terminal portion) that is blocked from engaging MIS12C.

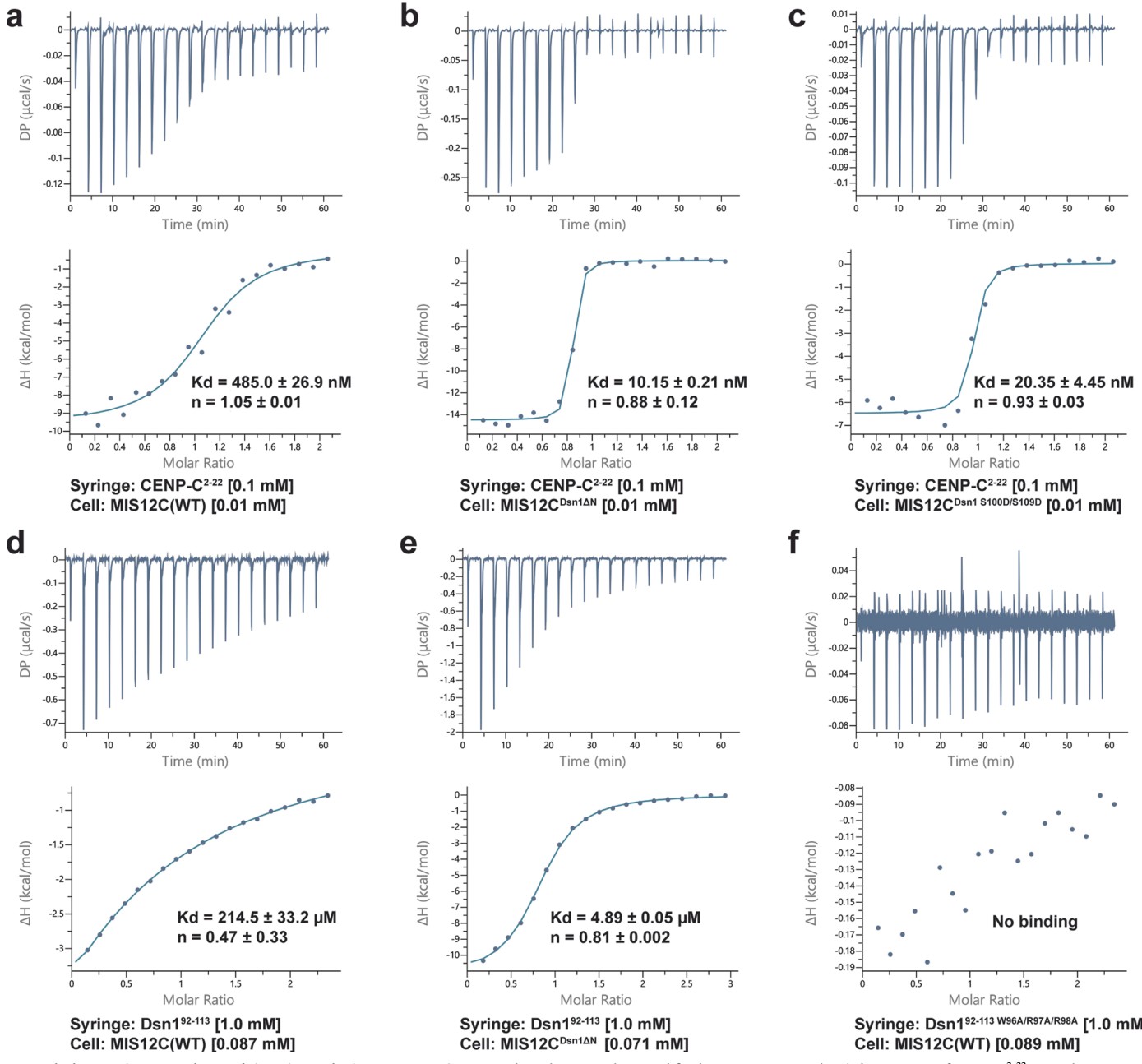

**Extended Data Fig. 4 | Isothermal titration calorimetry experiments.** The Kd and stoichiometries (n) values are averages between at least two experiments. The reported error values are calculated standard deviations. CENP-C$^{2-22}$ and Dsn1$^{92-113}$ peptides were injected using a syringe into the cell containing MIS12C protein variants. Protein and peptide concentrations in cell and syringe are indicated in square brackets. **a** Titration of CENP-C$^{2-22}$ peptide to full-length and unmodified MIS12C, MIS12C (WT). **b** Titration of CENP-C$^{2-22}$ peptide to MIS12C with deleted Dsn1$^N$ (6–113 residues deleted, MIS12C$^{Dsn1ΔN}$). **c** Titration of CENP-C$^{2-22}$ peptide to MIS12C with S100D and S109D substitutions in Dsn1 protein (MIS12C$^{Dsn1 S100D/S109D}$). **d** Titration of Dsn1$^{92-113}$ peptide to MIS12C (WT). **e** Titration of Dsn1$^{92-113}$ peptide to MIS12C$^{Dsn1ΔN}$. **f** Titration of Dsn1$^{92-113}$ peptide with W96A/R97A/R98A substitutions (Dsn1$^{92-113 W96A/R97A/R98A}$) to MIS12C (WT).

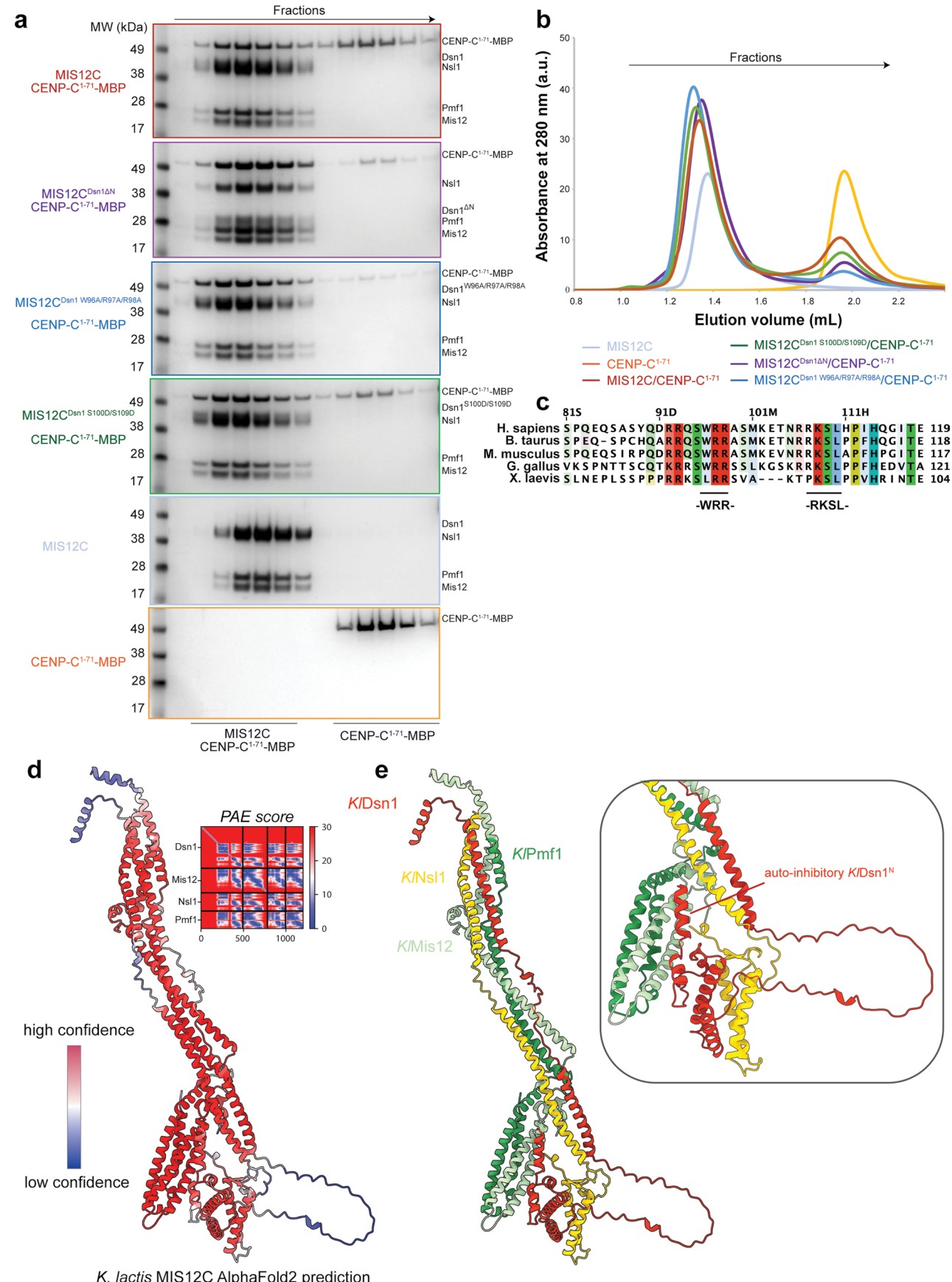

*K. lactis* MIS12C AlphaFold2 prediction

**Extended Data Fig. 5 | See next page for caption.**

**Extended Data Fig. 5 | Biochemical analysis of CENP-C interactions with MIS12C. a** Coomassie blue-stained SDS-PAGE gels of the MIS12C:CENP-C[1–71] interaction reconstitution. Wild-type MIS12C, MIS12C[Dsn1ΔN], MIS12C with W96A/R97A/R98A substitutions in Dsn1 (MIS12C[Dsn1W96A/R97A/R98A]) or MIS12C[Dsn1 S100D/S109D] was used in these experiments to test interaction with CENP-C[1–71] tagged at the C-terminus with Maltose Binding Protein (MBP, CENP-C[1–71]-MBP) to allow robust visualization of CENP-C. The binding experiments were repeated at least two independent times with identical results. **b** SEC elution chromatograms of all of the experiments described in (a). **c** Multiple sequence alignment of Dsn1 protein using model eukaryotes shows that the -WRR- linchpin and -RKSL- motif are conserved. **d** AlphaFold2 Multimer prediction of the *K. lactis* MIS12C structure. Coloured by pLDDT score. **e** AlphaFold2 Multimer prediction of the *K. lactis* MIS12C coloured by subunit with insert highlighting conservation of the auto-inhibitory Dsn1[N] mechanism.

**a**

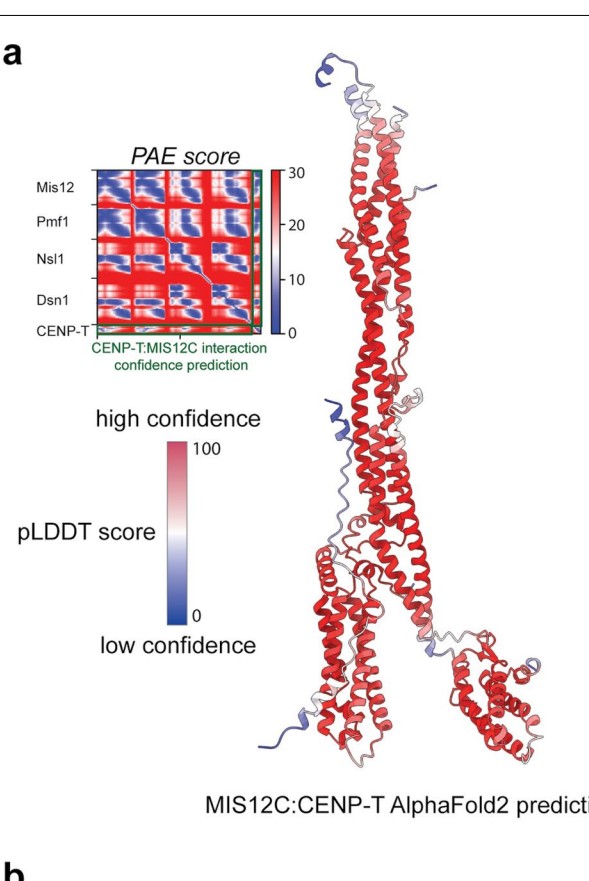

MIS12C:CENP-T AlphaFold2 prediction

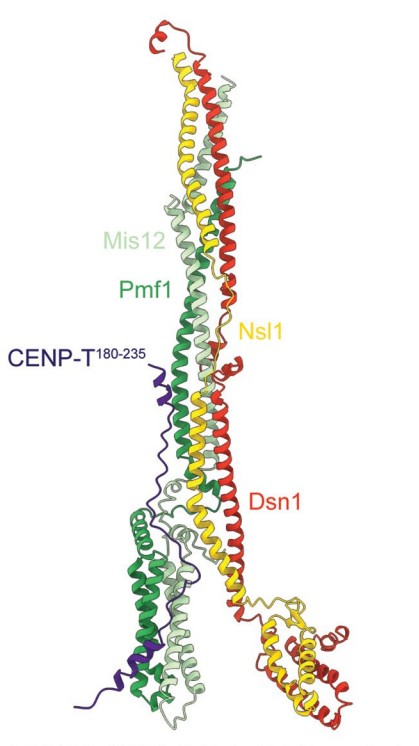

MIS12C:CENP-T AlphaFold2 prediction

**b**

**c**

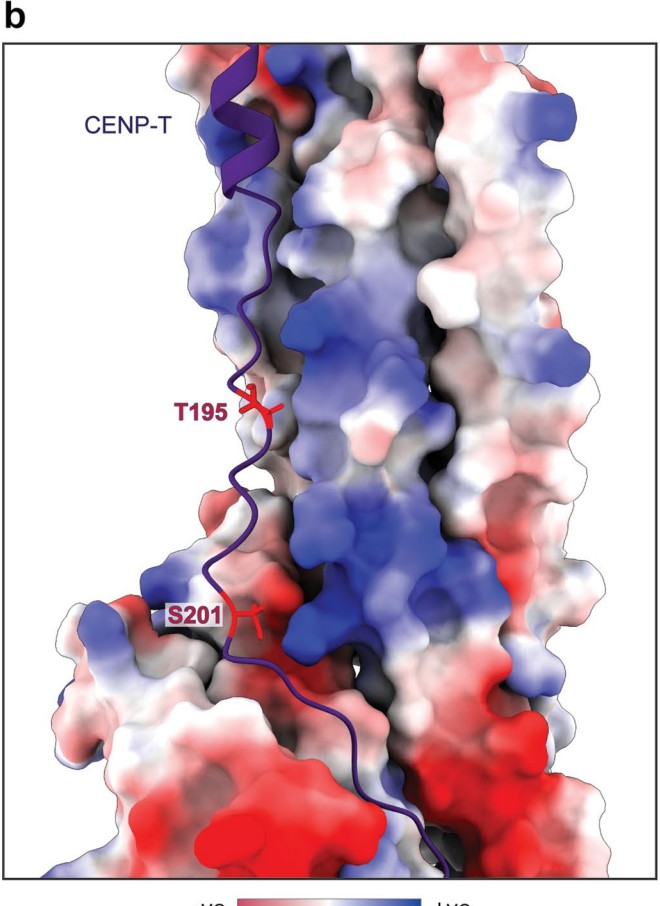

**Extended Data Fig. 6 | See next page for caption.**

**Extended Data Fig. 6 | AlphaFold2 model of CENP-T:MIS12C interactions.**
**a** AlphaFold2 Multimer prediction of the CENP-T:MIS12C interaction based on the full-length MIS12C protein sequences for all proteins apart from Dsn1, for which residues 113–356 were used to remove auto-inhibitory Dsn1[N] region. Only CENP-T residues 180–230 were used for this prediction. The model on the left is coloured by pLDDT score whereas the model on the right is coloured by chain. The residues forming the dominant interaction of CENP-T with MIS12 head 1 have pLDDT scores of 70–83. The positional alignment error (PAE) score is shown in the top left of the panel, with the CENP-T-MIS12C interaction confidence prediction indicated in green boxes. The blue-colouring indicates a low PAE score (high confidence). **b** A CENP-T-interacting region of MIS12C is shown as an electrostatic surface potential while Thr195 and Ser201 of CENP-T are highlighted in red. Both Thr195 and Ser201 are targets of CDK kinase in cells and both residues face positively-charged patches on MIS12C. **c** Overlay of CENP-T:MIS12C AlphaFold2 prediction on the MIS12C structure determined in this study. The auto-inhibitory Dsn1[N] region and CENP-T peptidic region are shown in space-filling representation, highlighting an extensive steric clash.

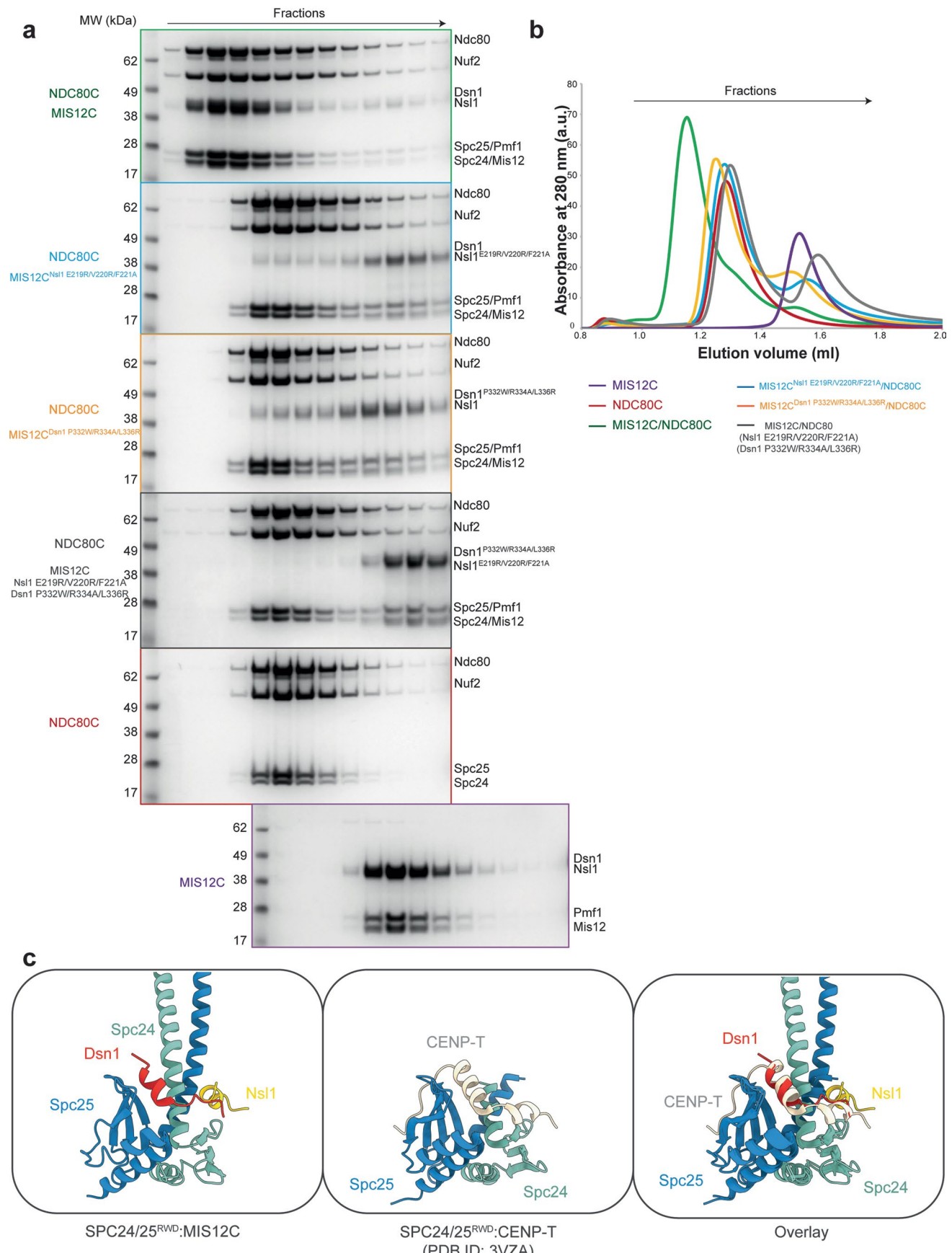

**a**

MW (kDa)

Fractions →

NDC80C
MIS12C

NDC80C
MIS12C^Nsl1 E219R/V220R/F221A

NDC80C
MIS12C^Dsn1 P332W/R334A/L336R

NDC80C
MIS12C
Nsl1 E219R/V220R/F221A
Dsn1 P332W/R334A/L336R

NDC80C

MIS12C

**b**

MIS12C
NDC80C
MIS12C/NDC80C
MIS12C^Nsl1 E219R/V220R/F221A/NDC80C
MIS12C^Dsn1 P332W/R334A/L336R/NDC80C
MIS12C/NDC80
(Nsl1 E219R/V220R/F221A)
(Dsn1 P332W/R334A/L336R)

**c**

SPC24/25^RWD:MIS12C

SPC24/25^RWD:CENP-T
(PDB ID: 3VZA)

Overlay

**Extended Data Fig. 7 | See next page for caption.**

**Extended Data Fig. 7 | Biochemical characterization of MIS12C:NDC80C interactions. a** Coomassie blue-stained SDS-PAGE gels of the MIS12C:NDC80C interaction reconstitutions. MIS12C (WT), MIS12C with E219/RV220R/F221A substitutions in Nsl1 (MIS12C$^{Nsl1 E219/RV220R/F221A}$) and MIS12C with P332W/R334A/L336R substitutions in Dsn1 (MIS12C$^{Dsn1 P332W/R334A/L336R}$) were tested for binding to full-length, unmodified NDC80C. The binding experiments were repeated at least two independent times with identical results. The gels are aligned according to the fraction number of the protein sample eluted from the size exclusion column. **b** SEC elution chromatograms of all of the experiments described in (a). **c.** Comparison of the Spc24:Spc25 interaction with peptides from MIS12C and CENP-T (PDB ID: 3VZA ref. 29).

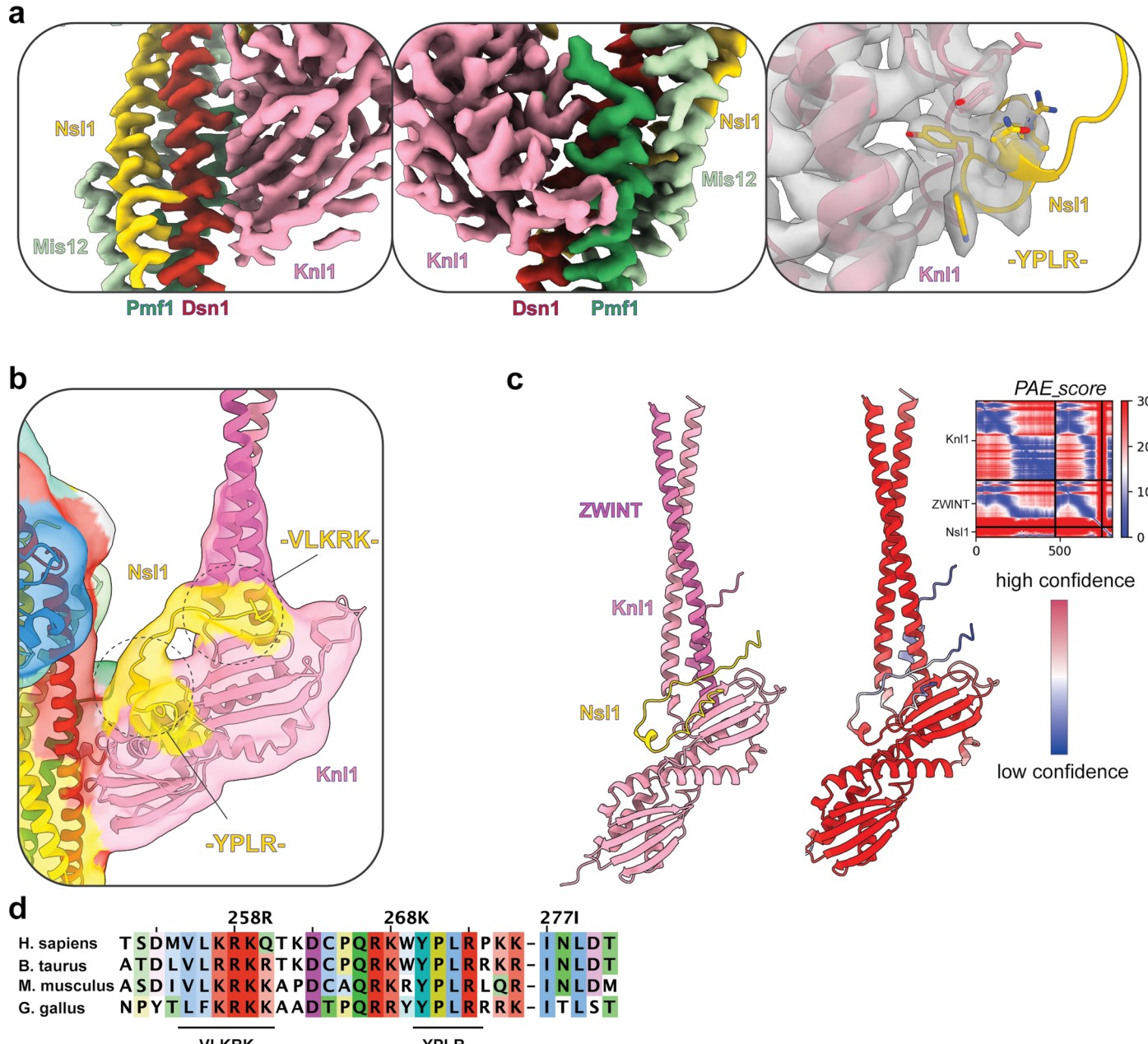

**Extended Data Fig. 8 | Molecular details of MIS12C:KNL1C interactions.**
**a** Cryo-EM density map of the KNL1C:MIS12C interface, highlighting the
resolution of interactions in this region. **b** Unsharpened and unmasked cryo-EM
map of the KMN network shows a long peptide of Nsl1 binding to the Knl1:ZWINT
interface and then folding across the Knl1 surface. **c** AlphaFold2 structure
prediction of the Knl1:ZWINT:Nsl1 complex, demonstrating strong agreement
with the experimental density maps in (b). The model on the left is coloured
by chain while the model on the right is coloured by pLDDT score. **d** Sequence
alignment of the Nsl1 protein showing conservation of two Nsl1 motifs involved
in Knl1 interaction.

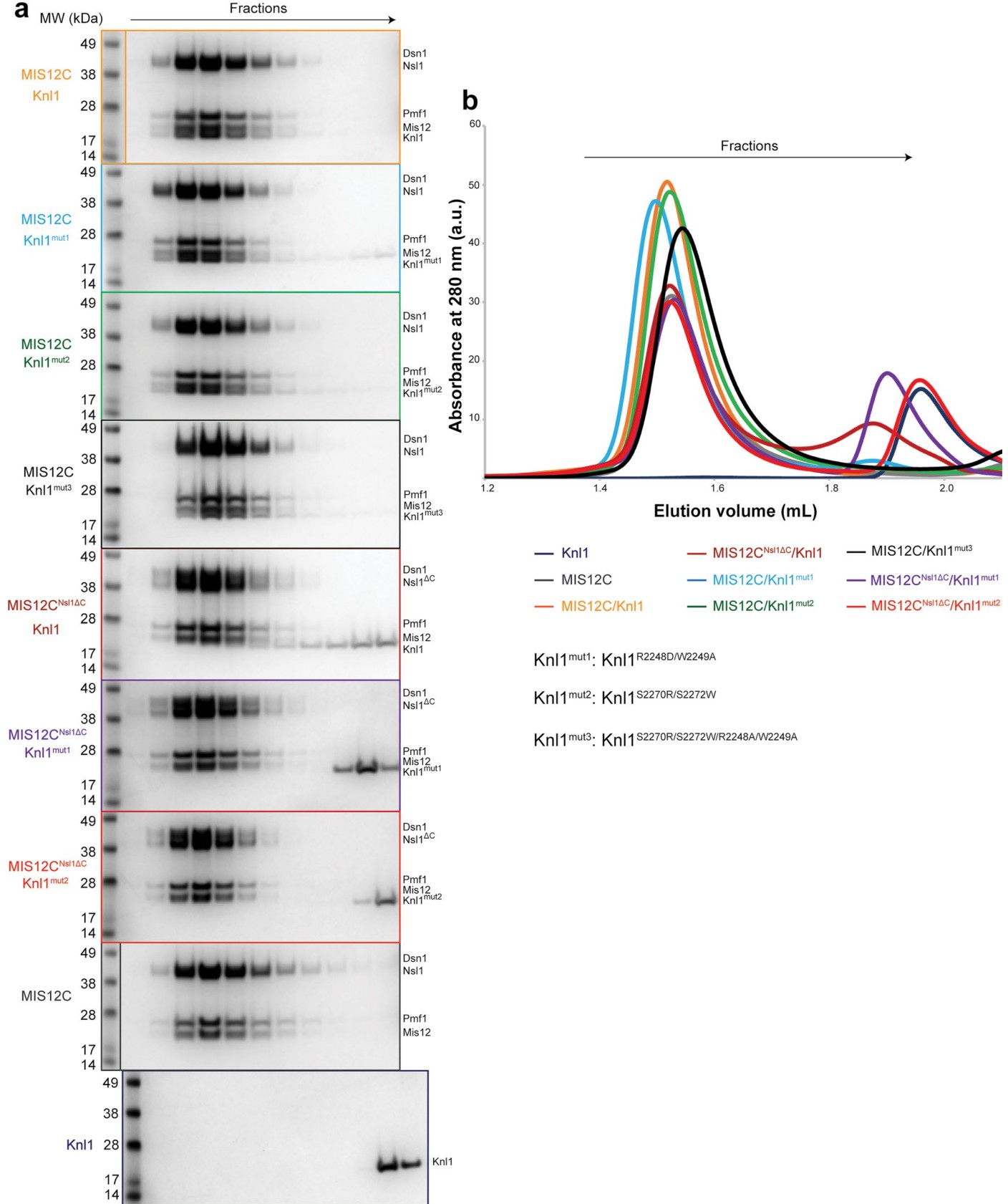

**Extended Data Fig. 9 | Biochemical validation of the MIS12C:Knl1 interface.**
**a** Coomassie blue-stained SDS-PAGE gels of the MIS12C:Knl1 interaction
reconstitutions. MIS12C (WT) and MIS12C with Nsl1 truncated at the C-terminus
(1–264 Nsl1 construct, MIS12C$^{Nsl1\Delta C}$) were tested in their ability to bind either
wild-type RWD domains of Knl1 (2131–2337 construct, labelled Knl1 for simplicity)
or Knl1 mutants (R2248D/W2249S substitutions, Knl1$^{mut1}$, and S2270R/S2272W

substitutions, Knl1$^{mut2}$, as well as S2270R/S2272W/R2248A/W2249S substitutions,
Knl1$^{mut3}$). The binding experiments were repeated at least two independent times
with identical results. The gels are aligned according to the fraction number
of the protein sample eluted from the size exclusion column. **b** SEC elution
chromatograms of all of the experiments described in (a).

David Barford

# Reporting Summary

## Statistics

For all statistical analyses, confirm that the following items are present in the figure legend, table legend, main text, or Methods section.

| n/a | Confirmed | |
|---|---|---|
| ☐ | ☒ | The exact sample size (*n*) for each experimental group/condition, given as a discrete number and unit of measurement |
| ☐ | ☒ | A statement on whether measurements were taken from distinct samples or whether the same sample was measured repeatedly |
| ☒ | ☐ | The statistical test(s) used AND whether they are one- or two-sided<br>*Only common tests should be described solely by name; describe more complex techniques in the Methods section.* |
| ☒ | ☐ | A description of all covariates tested |
| ☒ | ☐ | A description of any assumptions or corrections, such as tests of normality and adjustment for multiple comparisons |
| ☒ | ☐ | A full description of the statistical parameters including central tendency (e.g. means) or other basic estimates (e.g. regression coefficient) AND variation (e.g. standard deviation) or associated estimates of uncertainty (e.g. confidence intervals) |
| ☒ | ☐ | For null hypothesis testing, the test statistic (e.g. *F*, *t*, *r*) with confidence intervals, effect sizes, degrees of freedom and *P* value noted<br>*Give P values as exact values whenever suitable.* |
| ☒ | ☐ | For Bayesian analysis, information on the choice of priors and Markov chain Monte Carlo settings |
| ☒ | ☐ | For hierarchical and complex designs, identification of the appropriate level for tests and full reporting of outcomes |
| ☒ | ☐ | Estimates of effect sizes (e.g. Cohen's *d*, Pearson's *r*), indicating how they were calculated |

*Our web collection on statistics for biologists contains articles on many of the points above.*

## Software and code

Policy information about availability of computer code

| Data collection | EM data collection Commercial software was used: EPU **(version 3.4.0.5704 REL)** from Thermo Fisher Scientific for automated cryo-EM data collection. |
|---|---|
| Data analysis | Micrograph movie frames were aligned with MotionCor2. CTF estimation was performed using CTFFIND4 v4.1.14 . Data were processed using RELION v4.0 and cryoSPARC v2.14.2. Particle picking was performed with TOPAZ **v0.2.5**. Models were built using AlphaFold (https://alphafold.ebi.ac.uk/), Chimera X v1.4 and COOT v0.8.9.2 and real-space refinement in PHENIX **v1.20.1**. Model validation was performed using MolProbity v4.5. Visualization was done using Chimera X v1.4. **Sequence alignments were performed using Uniprot, Clutal Omega (https://** www.ebi.ac.uk/Tools/msa/clustalo/) and displayed with JALVIEW v 1.0. Structure prediction was performed **using local instalation of** the ALPHAFOLD2 software. ITC data was analysed using MicroCal PEAQ-ITC Analysis Software 1.0.0.1258. |

For manuscripts utilizing custom algorithms or software that are central to the research but not yet described in published literature, software must be made available to editors and reviewers. We strongly encourage code deposition in a community repository (e.g. GitHub). See the Nature Portfolio guidelines for submitting code & software for further information.

## Data

Policy information about availability of data

All manuscripts must include a data availability statement. This statement should provide the following information, where applicable:

- Accession codes, unique identifiers, or web links for publicly available datasets
- A description of any restrictions on data availability
- For clinical datasets or third party data, please ensure that the statement adheres to our policy

The cryoEM map has been deposited with the Electron Microscopy Data Bank (EMDB) with accession code: EMD-17814 Protein coordinate has been deposited with RCSB with accession code: 8PPR. Publicly available PDB entries used in this study are: 5LSK, 3VZA and AlphaFold2 AF-Q8NG31-F1. CENP-C uniprot entries used for sequence alignments: Q03188 (H. sapiens), P49452 (M. musculus), O57392 ( G. gallus), Q66LI3 (B. taurus); DSN1 uniprot entries used for sequence alignment: Q9H410 (H. sapiens), A6QQ81 (B. taurus), Q9CYC5 (M. musculus), Q1T768 (G. gallus), A0A8J0US97 (X. laevis).

## Research involving human participants, their data, or biological material

Policy information about studies with human participants or human data. See also policy information about sex, gender (identity/presentation), and sexual orientation and race, ethnicity and racism.

| | |
|---|---|
| Reporting on sex and gender | N/A |
| Reporting on race, ethnicity, or other socially relevant groupings | N/A |
| Population characteristics | N/A |
| Recruitment | N/A |
| Ethics oversight | N/A                                                    c |

Note that full information on the approval of the study protocol must also be provided in the manuscript.

# Field-specific reporting

Please select the one below that is the best fit for your research. If you are not sure, read the appropriate sections before making your selection.

☒ Life sciences ☐ Behavioural & social sciences ☐ Ecological, evolutionary & environmental sciences

For a reference copy of the document with all sections, see nature.com/documents/nr-reporting-summary-flat.pdf

# Life sciences study design

All studies must disclose on these points even when the disclosure is negative.

| | |
|---|---|
| Sample size | No statistical method was used for sample size calculation. For the KMN network complex, a total of 17,815 micrographs were collected. A total number of 2,549,620 particles were auto-picked, of which 599,831 were used for a final reconstruction. For cryoEM reconstructions, sample sizes were determined by available electron microscopy time and the number of particles on each micrograph obtained during the collection time. |
| Data exclusions | Through 2D and 3D classification procedures, broken particles or particles that do not belong to the classes of interests were discarded. This is a standard practice in cryoEM studies to obtain homogeneous cryoEM structures. |
| Replication | CryoEM datasets were collected with multiple samples in separate imaging sessions. All biochemical experiments were repeated at least in three independent experiments and are all reproducible. |
| Randomization | No randomization was performed, since this study did not allocate experimental groups. |
| Blinding | No blinding was performed, since it is not relevant to this study. |

# Reporting for specific materials, systems and methods

We require information from authors about some types of materials, experimental systems and methods used in many studies. Here, indicate whether each material, system or method listed is relevant to your study. If you are not sure if a list item applies to your research, read the appropriate section before selecting a response.

## Materials & experimental systems

| n/a | Involved in the study |
|-----|----------------------|
| ☒ | ☐ Antibodies |
| ☐ | ☒ Eukaryotic cell lines |
| ☒ | ☐ Palaeontology and archaeology |
| ☒ | ☐ Animals and other organisms |
| ☒ | ☐ Clinical data |
| ☒ | ☐ Dual use research of concern |
| ☒ | ☐ Plants |

## Methods

| n/a | Involved in the study |
|-----|----------------------|
| ☒ | ☐ ChIP-seq |
| ☒ | ☐ Flow cytometry |
| ☒ | ☐ MRI-based neuroimaging |

## Eukaryotic cell lines

Policy information about cell lines and Sex and Gender in Research

| | |
|---|---|
| Cell line source(s) | High-5 insect cell (BTI-TN-5B1-4) were obtained from ThermoFisher Scientific. |
| Authentication | The High-5 insect cell line was not authenticated |
| Mycoplasma contamination | The High-5 insect cell line was not tested for mycoplasma contamination. |
| Commonly misidentified lines (See ICLAC register) | None |

