## [Peer Review File · Nature Structural & Molecular Biology]

Peer Review Information

Manuscript Title: Structure of the human outer kinetochore KMN network complex

Corresponding author name(s): Stanislau Yatskevich, David Barford

Reviewer Comments & Decisions:

Decision Letter, initial version:

Message: 11th Sep 2023

Dear Dr. Barford,

Thank you again for submitting your manuscript "Structure of the human outer kinetochore KMN network complex". I apologise for the delay in responding, which resulted from the difficulty in obtaining suitable referee reports. Nevertheless, we now have comments (below) from the 2 reviewers who evaluated your paper. In light of these reports, we remain interested in your study and would like to see your response to the comments of the referees, in the form of a revised manuscript.

You will see that though both experts appreciate the potential of the structural findings and their implications, they raise important concerns, at both the technical but also mechanistic level, that need to be convincingly addressed in a revised manuscript. More specifically, reviewer #2 brings up technical concerns with respect to the analysis of the cryo-EM data that must be addressed. In addition, reviewer #2 provides important guidance with respect to additional biochemical experiments, ITC and mutational analyses to substantiate drawn conclusions. Finally, reviewer #1 has put forward extensive guidelines with respect to rephrasing to accurately describe and contextualise the novel findings, and citing the relevant literature - please revise the text carefully, to avoid misunderstandings.

Please be sure to address/respond to all concerns of the referees in full in a point-by-point response and highlight all changes in the revised manuscript text file. If you have comments that are intended for editors only, please include those in a separate cover letter.

We expect to see your revised manuscript within 3-6 months. If you cannot send it within this time, please contact us to discuss an extension; we would still consider your revision, provided that no similar work has been accepted for publication at NSMB or published elsewhere.

Reporting Summary:

When submitting the revised version of your manuscript, please pay close attention to our [href="https://www.nature.com/nature-portfolio/editorial-policies/image-integrity">Digital Image Integrity Guidelines. and to the following points below:](https://www.nature.com/nature-portfolio/editorial-policies/image-integrity)

Data availability: this journal strongly supports public availability of data. All data used in accepted papers should be available via a public data repository, or alternatively, as

Supplementary Information. If data can only be shared on request, please explain why in your Data Availability Statement, and also in the correspondence with your editor. Please note that for some data types, deposition in a public repository is mandatory - more information on our data deposition policies and available repositories can be found below: <https://www.nature.com/nature-research/editorial-policies/reporting-standards#availability-of-data>

Nature Structural & Molecular Biology is committed to improving transparency in authorship. As part of our efforts in this direction, we are now requesting that all authors identified as 'corresponding author' on published papers create and link their Open Researcher and Contributor Identifier (ORCID) with their account on the Manuscript Tracking System (MTS), prior to acceptance. This applies to primary research papers only. ORCID helps the scientific community achieve unambiguous attribution of all scholarly contributions. You can create and link your ORCID from the home page of the MTS by clicking on 'Modify my Springer Nature account'. For more information please visit [visit www.springernature.com/orcid](http://www.springernature.com/orcid).

[redacted]

Sincerely,

Dimitris Typas
Associate Editor
Nature Structural & Molecular Biology
ORCID: 0000-0002-8737-1319

Referee expertise:

Referee #1: Structural biology of kinetochores

Referee #2: Kinetochores structure and function

Reviewers' Comments:

Reviewer #1:

Remarks to the Author:

Comments on Yatskevich et al

This paper reports structural and biochemical analysis of the so-called KMN network, a component of the microtubule attachment apparatus of kinetochores. (It's not a network - as the structure shows it's an elongated, branched complex -- but we're probably stuck with the nomenclature.) It builds on many results from other groups, accumulated over the past 20 or more years, from both genetic analyses (especially in yeast), biochemical dissections and reconstitutions, and structural analyses. There are three principal new results from the structure. (1) The interface between Spc24:25 and Mis12c is relatively rigid, imparting a defined directionality between Mis12c and the proximal end of Ndc80c. (2) The equally well defined interface between Knl1 and Mis12c shows that the Knl1:ZWINT heterodimer projects "upward", parallel (or nearly so) to the coil-coil of Spc24:25. (3) The results also give a direct structural interpretation to the previously described autoinhibition of the yeast Mis12c equivalent and its alleviation by phosphorylation of serine residues by Ipl1 as well as of the direct competition between the Dsn1 N-terminal segment and the N-terminal segment of CENP-C (Mif2) (likewise previously documented biochemically for the yeast orthologues).

The structural results are fully consistent with those in the co-submitted paper from Musacchio's group and includes some pieces (the ZWINT:Knl1 interaction) not seen in the latter, while the Dortmund paper adds important experiments in cells to validate functional relevance.

There are some exaggerations and misunderstandings that need correction, and some missing qualifications. The points below are listed sequentially in their order in the MS, rather in in order of significance -- some are trivial editorial remarks, others are more substantive.

Abstract: (1) Does the KMN really withstand particularly strong forces? If we assume that there are 20 microtubules attached to a human kinetochore (Meraldi) and that each one has six Mis12-based connections through Ndc80, then estimates of up to 700 pN per kinetochore (Nicklas' top estimate for grasshopper kinetochores) comes down to ~5 pN per connection, even neglecting Cnn1-Ndc80 connections and possibly cross-connections from Ska. The key issue, as Biggins and Murray showed many years ago, is sensing tension (by Aurora B), not withstanding it. (2) The last sentence is a speculation, as there is no direct evidence for this particular restriction. Speculation is appropriate in a Discussion (when carefully qualified as such), but not in an Abstract. They should write "suggest how selectively relieving this autoinhibition ... might restrict"

Introduction

First paragraph: see note above about force.

Second paragraph: "projects outwards" - ""projects laterally" might be better -- on first reading, I interpreted "outwards" as "toward the C-terminal end of the Mis12 complex".

Composite interface section:

Yeast Dsn1 binds the same site as CENP-T (PDB 5T6J), so the last paragraph needs some modification.

Overall model section:

Whether the hinge bends as illustrated in Fig. 5 is not (in this reviewer's view) well established. There are, in the rotary shadowed images of Huis in't Velt et al, some strong bends, but spraying onto mica (the procedure in rotary shadowing) can generate strong distortions, even in fairly rigid coiled-coils. It is clear that there is a kink or hinge at the position inferred from measurements in that paper, but the extreme bend is still speculative. It would be better to modify both text and Fig. 5 to emphasize that the bend is not something their data show -- many readers will just look at pictures, not at explanations. Even a question mark by the bend would help.

Discussion. Overall, the phrasing does a good job of claiming more for this work than really justified. For example, in my comment on the Abstract, I pointed out that there is no experimental evidence (to my knowledge) that "initial weak binding ... allows KMN to be recruited" but that "only KMN bound to the functional centromeres would be stabilized and converted" It is a perfectly reasonable speculation, but the way it is stated claims too much, especially as the speculation builds on results from many other groups and does not really depend on the structure, just on the autoinhibition and the strength of binding. There is still an argument in the literature about whether Aurora B needs to be kinetochore tethered in order to phosphorylate at least some of its various targets (see, for example, a recent paper from Biggins).

Fig. 1. The 2D class averages are very pretty.

Fig. 2. Inconsistency of colours from panel to panel makes the figure very hard to understand. The colours in panels a and b should carry forward to the rest of the figure. In the superposition in e, one can use white or pale gray for the open (activated) model, but please use red for Dsn1, etc. as in a. The N-terminal segment can, of course, be a different shade of red -- e.g., rose or even pink -- to highlight it, while still keeping it clearly identified.

Fig. 2, panel g. There's something dodgy about Kd for Mis12c(deltaN) -- $1 \text{ pM} \pm 3 \text{ nM}$ just doesn't make sense. The binding might have been irreversible, so giving an apparently tiny Kd, but unless the system is an equilibrium, the assumptions of measuring a Kd are wrong.

Reviewer #2:

Remarks to the Author:

The kinetochore is a large protein complex on the centromere region for accurate chromosome segregation. The kinetochore is divided into two major groups: Constitutive Centromere Associated Network (CCAN) and Knl1, Mis12, and Ndc80 complexes (KMN network). CCAN complex is associated with centromeric chromatin and constitutively

localizes to the centromere region. KMN network is recruited on CCAN during mitosis and connects with spindle microtubules via microtubule binding activity of the Ndc80 complex. Recent structural analyses on the reconstituted kinetochore (sub)complexes gave us many insights for the kinetochore architecture. In addition to the CCAN structure, structures of various parts of the KMN network have been solved. However, entire structure of the KMN network was still unclear, and it is an interesting issue to clarify how each subcomplex assembles to form the entire KMN network.

In this study, Yatskevich et al. reconstituted the entire KMN network in vitro and applied it to cryo-EM analysis. They combined AlphaFold2 prediction with cryo-EM data and proposed a structural model for the entire KMN network. They observed an auto-inhibited form of the Mis12 complex, which inhibits to bind to CCAN, and this form is suppressed by phosphorylation of Dsn1 in the Mis12 complex.

Overall, the structural model is consistent with another work by Musacchio and colleagues, which was also submitted to same Journal, and provides some new interaction surfaces in the entire KMN complex. On the other hand, many results are predictable, based on previously published structures. This reviewer has some concerns authors should address.

Cryo-EM analysis

My major concern is on criteria of the cryo-EM structure model building in this study. According to a PDB validation report (9. Map-model fit), a deposited model, probably the entire model of the KMN junction core shown in Fig 1b, contains regions which are unmapped by cryo-EM, experimentally. I wonder whether this way fits the criteria of model building and refinement using experimental cryo-EM maps. From this viewpoint, in this manuscript, descriptions and presentations of the cryo-EM structure appear to be overstated and misleading. My questions and comments related to this issue are listed below:

- 1) According to Extended Data Fig 2, the map for the head domains of Mis12C is ambiguous in the 3.0 Å resolution cryo-EM map, and further refinement with a mask improved the map for the Mis12C heads, which resolution is 3.4 Å. In Extended Table1, the model resolution is 2.7 Å. Please clarify which map is presented in Fig 1a and was used for model building and refinement explained in Methods. The cryo-EM reconstitution demonstrated here seems to be 3.4 Å resolution although resolution of the reconstitution is described to be 3.0 Å in a main text.
- 2) The model of the KMN junction shown in Fig 1b is generally expected to be consistent with the cryo-EM map in Fig 1a. I understood that this map was used for structure determination. It is misleading to include the model of long coiled-coil regions which experimental map was not shown in Fig 1b.
- 3) The models of coiled-coil regions of Spc24/25 and KNL1/ZWINT were built based on the map with a lower threshold and 2D class averaged images, but the map shown in Fig 2d appears to be too ambiguous and the model was built beyond it. This part of the structure is critical for authors' statement about parallel arrangements of two coiled-coil regions. A map presentation showing model fitting quality is necessary. What is local resolution of the map for these coiled-coil regions? Is it high enough at least to identify each helix? Is it possible to improve the map resolution using the same approach applied to the Mis12C head region? If experimental data is not qualified enough, this structural feature should be described as a hypothetical/predicted model in "Discussion", and then Figs 1d and 1e could be accepted as supportive data.
- 4) According to description in Methods (Cryo-EM model building and refinement),

AlphaFold2 models of NDC80C and KNL1 were used as initial models, but in Extended Table 1, 'initial model used' is 'ab initio'. Please provide more accurate explanation about initial models used for the structural determination. Was the crystal structure of Mis12C used as an initial model?

Biochemical analysis and others

- 1) The authors propose that Dsn1 N-terminus inhibits CENP-C-Mis12C interaction (Fig 2g). To support this proposal, authors performed ITC experiments using CENP-C1-71 and Dsn192-103 peptides to evaluate their binding to Mis12CDsn1ΔN. However, CENP-C1-71 binds Mis12CDsn1ΔN much more strongly than Dsn192-103 (Kd of Mis12CDsn1ΔN -CENP-C1-71 interaction was 0.001 nM, while Kd of Mis12CDsn1ΔN-Dsn192-103 was 4.92 μM). This may be caused by using of unappropriated peptides for these ITC experiments, because CENP-C1-71 and Dsn192-103 peptides do not appear to be comparable for their Mis12CDsn1ΔN binding. In the Mis12C auto-inhibited state structure authors observed, CENP-C6-22 is precluded from Mis12C-binding, while CENP-C24-48 can still interact with Mis12C. To evaluate the auto-inhibition model using ITC, the longer Dsn1 peptide containing both WRR linchpin and RKSL motif and the CENP-C6-22 peptide should be used. Binding of CENP-C6-22 to Mis12CDsn1ΔN should be compared with that of the longer Dsn1 to Mis12CDsn1ΔN.
- 2) In Extended Data Fig 5c and d, the authors explain that the auto-inhibition mechanism by Mis12C is likely to be evolutionarily conserved, based on *K. lactis* MIND complex structure predicted by AlphaFold2. However, as the authors mentioned, AlphaFold2 failed to predict the structure of the *S. cerevisiae* MIND complex. Therefore, it is hard to conclude evolutionary conservation by only the *K. lactis* MIND complex prediction. To strengthen this hypothesis, the authors should provide the sequence alignment of Dsn1 N-terminus across various species and assess whether WRR linchpin and RKSL motif are evolutionarily conserved.
- 3) In the section titled "Dsn1N likely inhibits the binding of the CENP-T to Mis12C", the authors propose the regulatory mechanisms of the CENP-T-Mis12C interaction, mediated by the Dsn1 N-terminus. This model has already proposed in previous studies, as the authors mentioned, and no new experimental data about CENP-T-Mis12C interaction is demonstrated in this analysis. This paragraph should be removed, and the essence of the proposal based on the predicted model for the CENP-T-Mis12C complex should be shortly described in the Discussion section.
- 4) In Extended Data Fig 9, the authors conclude that Knl1mut1 and Knl1mut2 had a modest effect on reducing the affinity of the Knl1-Mis12C interaction. However, in the SEC elution chromatograms (Extended data Fig 9b), Knl1mut2 doesn't appear to affect the Knl1-Mis12C interaction. Please show more convincing data to demonstrate the effect of Knl1mut2. Otherwise, it is hard to conclude that Knl1mut2 reduced binding to Mis12C.
- 5) Based on the results from Extended Data Fig 9, the authors conclude that Knl1RWD-N dominates Knl1-Mis12C interactions. This conclusion might not be strictly correct. To disrupt the Knl1RWD-N-Mis12C interaction, Nsl1 C-terminus associated with this interaction was removed. On the other hand, to disrupt the Knl1RWD-C-Mis12C interaction, mutations were introduced in only one of the two important regions of Knl1. Please introduce mutations in both regions involved in the binding with Mis12C in Knl1RWD-C (Knl1R2248D/W2249A/S2270R/S2272W) and investigate the binding with Mis12C, alternatively please change to the correct statement.

Minor comments

- 1) Please show the N- and C-terminus of each protein in Fig 1b for clarity.
- 2) In Fig 2a and b, "KSL" should be "RKSL".

- 3) It is difficult to recognize the hydrophobic pockets of Mis12 and Pmf1 in Fig 2. Please provide an improved structural figure showing the hydrophobic pockets for instance using a surface presentation.
- 4) In Fig 2f, please indicate the locations of His8, Asn11, Arg14, and Arg15 of CENP-C.
- 5) The deleted region (amino acid residue numbers) in Dsn1 Δ N (p5) or Nsl1 Δ C (p8) should be described in the main text.
- 6) In the graph shown PAE score (Extended Data Fig 5c, Extended Data Fig 6a, Extended Data Fig 8c), please indicate which protein is corresponding to A, B, C, or D in Y-axis.
- 7) In p6 lower part, "MIS12C complex" should be "MIS12C".
- 8) Please correct the numbering of the figure legend of Extended Data Fig7 (d, e, f -> a, b, c)
- 9) The head domains in carton schematic in Fig 1c and Fig 5 look like formed by only Pmf1 (green) and Dsn1 (red) only.

Author Rebuttal to Initial comments

Reviewer #1:

Remarks to the Author:

This paper reports structural and biochemical analysis of the so-called KMN network, a component of the microtubule attachment apparatus of kinetochores. (It's not a network -- as the structure shows it's an elongated, branched complex -- but we're probably stuck with the nomenclature.) It builds on many results from other groups, accumulated over the past 20 or more years, from both genetic analyses (especially in yeast), biochemical dissections and reconstitutions, and structural analyses. There are three principal new results from the structure. (1) The interface between Spc24:25 and Mis12c is relatively rigid, imparting a defined directionality between Mis12c and the proximal end of Ndc80c. (2) The equally well defined interface between Knl1 and Mis12c shows that the Knl1:ZWINT heterodimer projects "upward", parallel (or nearly so) to the coiled-coil of Spc24:25. (3) The results also give a direct structural interpretation to the previously described autoinhibition of the yeast Mis12c equivalent and its alleviation by phosphorylation of serine residues by Ipl1 as well as of the direct competition between the Dsn1 N-terminal segment and the N-terminal segment of CENP-C (Mif2) (likewise previously documented biochemically for the yeast orthologues).

The structural results are fully consistent with those in the co-submitted paper from Musacchio's group and includes some pieces (the ZWINT:Knl1 interaction) not seen in the latter, while the Dortmund paper adds important experiments in cells to validate functional relevance.

There are some exaggerations and misunderstandings that need correction, and some missing qualifications. The points below are listed sequentially in their order in the MS, rather in in order of significance -- some are trivial editorial remarks, others are more substantive.

We thank the reviewer for carefully and critically reading the manuscript, and for suggesting important clarifications and corrections that have improved the manuscript.

Abstract: (1) Does the KMN really withstand particularly strong forces? If we assume that there are 20 microtubules attached to a human kinetochore (Meraldi) and that each one has six Mis12-based connections through Ndc80, then estimates of up to 700 pN per kinetochore (Nicklas' top estimate for grasshopper kinetochores) comes down to ~5 pN per connection, even neglecting Cnn1-Ndc80 connections and possibly cross-connections from Ska. The key issue, as Biggins and Murray showed many years ago, is sensing tension (by Aurora B), not withstanding it. (2) The last sentence is a speculation, as there is no direct evidence for this particular restriction. Speculation is appropriate in a Discussion (when carefully qualified as such), but not in an Abstract. They should write "suggest how selectively relieving this autoinhibition ... might restrict"

Thank you for highlighting these important biological implications.

1) Reference to force has been removed from the Abstract.

2) The last sentence was modified as suggested.

Introduction

First paragraph: see note above about force.

Reference to force has been removed from the text.

Second paragraph: "projects outwards" - "projects laterally" might be better -- on first reading, I interpreted "outwards" as "toward the C-terminal end of the Mis12 complex".

Thank you for this suggestion. Projects laterally is now used.

Composite interface section:

Yeast Dsn1 binds the same site as CENP-T (PDB 5T6J), so the last paragraph needs some modification.

We apologize for this omission. Reference to this structure is now included and the last paragraph of that section modified by deleting a sentence and by adding:

'These results are in agreement with a prior crystal structure of yeast Spc24:Spc25 in complex with a peptide modelled on the Spc24:Spc25-binding site of Dsn1 that showed Dsn1 and CENP-T share the same binding site on Spc24:Spc25¹⁰.' (page 7, lines 35-37).

Overall model section:

Whether the hinge bends as illustrated in Fig. 5 is not (in this reviewer's view) well established. There are, in the rotary shadowed images of Huis in't Velt et al, some strong bends, but spraying onto mica (the procedure in rotary shadowing) can generate strong distortions, even in fairly rigid coiled-coils. It is clear that there is a kink or hinge at the position inferred from measurements in that paper, but the extreme bend is still speculative. It would be better to modify both text and Fig. 5 to emphasize that the bend is not something their data show -- many

readers will just look at pictures, not at explanations. Even a question mark by the bend would help.

Thank you for these suggestions. A sentence has been added to the overall model section: 'However, we do not have experimental evidence to confidently state that the bent state exists in solution.' (page 9, lines 27-28). A question mark has been added to Fig. 5 next to the bent state, and explained in the figure legend.

Discussion. Overall, the phrasing does a good job of claiming more for this work than really justified. For example, in my comment on the Abstract, I pointed out that there is no experimental evidence (to my knowledge) that "initial weak binding ... allows KMN to be recruited" but that "only KMN bound to the functional centromeres would be stabilized and converted" It is a perfectly reasonable speculation, but the way it is stated claims too much, especially as the speculation builds on results from many other groups and does not really depend on the structure, just on the autoinhibition and the strength of binding. There is still an argument in the literature about whether Aurora B needs to be kinetochore tethered in order to phosphorylate at least some of its various targets (see, for example, a recent paper from Biggins).

A number of sentences have been deleted from the discussion, including: 'Therefore, our work provides a molecular explanation of how outer kinetochore recruitment is restricted specifically to centromeres with a functional Aurora B kinase.'

The remaining sentences have been rephrased to suggest a more speculative nature.

Fig. 1. The 2D class averages are very pretty.

Thank you.

Fig. 2. Inconsistency of colours from panel to panel makes the figure very hard to understand. The colours in panels a and b should carry forward to the rest of the figure. In the superposition in e, one can use white or pale gray for the open (activated) model, but please use red for Dsn1, etc. as in a. The N-terminal segment can, of course, be a different shade of red -- e.g., rose or even pink -- to highlight it, while still keeping it clearly identified.

Thank you for pointing out the necessary and important clarifications to the figure. The colour scheme has been modified to be consistent throughout the figure, including changing the colour scheme of panel e as suggested. We agree that this has improved the clarity of the figure.

Fig. 2, panel g. There's something dodgy about Kd for Mis12c(deltaN) -- $1 \text{ pM} \pm 3 \text{ nM}$ just doesn't

make sense. The binding might have been irreversible, so giving an apparently tiny K_d , but unless the system is a equilibrium, the assumptions of measuring a K_d are wrong.

Thank you for pointing this out. The reviewer makes an interesting point that the binding might be irreversible. We should have been aware that measuring pM affinities are beyond the range of reliable estimation using ITC. In response to referee 2 we have replaced the affinity data for CENP-C¹⁻⁷¹ binding to MIS12C (and mutants) with CENP-C²⁻²² (residues 2-22), the rationale being that this is the region of CENP-C blocked from binding to the auto-inhibited MIS12C. These new data show affinities ranging from low to high nM, with respectable error values.

Reviewer #2:

Remarks to the Author:

The kinetochore is a large protein complex on the centromere region for accurate chromosome segregation. The kinetochore is divided into two major groups: Constitutive Centromere Associated Network (CCAN) and Knl1, Mis12, and Ndc80 complexes (KMN network). CCAN complex is associated with centromeric chromatin and constitutively localizes to the centromere region. KMN network is recruited on CCAN during mitosis and connects with spindle microtubules via microtubule binding activity of the Ndc80 complex. Recent structural analyses on the reconstituted kinetochore (sub)complexes gave us many insights for the kinetochore architecture. In addition to the CCAN structure, structures of various parts of the KMN network have been solved. However, entire structure of the KMN network was still unclear, and it is an interesting issue to clarify how each subcomplex assembles to form the entire KMN network.

In this study, Yatskevich et al. reconstituted the entire KMN network in vitro and applied it to cryo-EM analysis. They combined AlphaFold2 prediction with cryo-EM data and proposed a structural model for the entire KMN network. They observed an auto-inhibited form of the Mis12 complex, which inhibits to bind to CCAN, and this form is suppressed by phosphorylation of Dsn1 in the Mis12 complex.

Overall, the structural model is consistent with another work by Musacchio and colleagues, which was also submitted to same Journal, and provides some new interaction surfaces in the entire KMN complex. On the other hand, many results are predictable, based on previously published structures. This reviewer has some concerns authors should address.

Cryo-EM analysis

My major concern is on criteria of the cryo-EM structure model building in this study. According to a PDB validation report (9. Map-model fit), a deposited model, probably the entire model of the KMN junction core shown in Fig 1b, contains regions which are unmapped by cryo-EM, experimentally. I wonder whether this way fits the criteria of model building and refinement using experimental cryo-EM maps. From this viewpoint, in this manuscript, descriptions and presentations of the cryo-EM structure appear to be overstated and misleading. My questions and

comments related to this issue are listed below:

1) According to Extended Data Fig 2, the map for the head domains of Mis12C is ambiguous in the 3.0 Å resolution cryo-EM map, and further refinement with a mask improved the map for the Mis12C heads, which resolution is 3.4 Å. In Extended Table 1, the model resolution is 2.7 Å. Please clarify which map is presented in Fig 1a and was used for model building and refinement explained in Methods. The cryo-EM reconstitution demonstrated here seems to be 3.4 Å resolution although resolution of the reconstitution is described to be 3.0 Å in a main text.

We thank the reviewer for carefully and critically reading the manuscript, and for suggesting additional analysis and experiments that have improved the manuscript.

The reviewer pointed out an incomplete description of the model building. Building of the large macromolecular models into cryo-EM maps frequently involves multiple cryo-EM refinements that are focused on different parts of the molecule, thereby generating numerous cryo-EM maps. Different maps are frequently used to accurately model different regions of the protein. In the previous manuscript, our description of which maps and coordinates are represented in figures and used for modelling and refinement was unclear.

To address this particular point, we modified Figure 1 figure legend to describe more precisely which map is shown:

- a. Composite cryo-EM density map of the human KMN^{Junction} complex comprised of the rigid Spc24:Spc25:MIS12C:Knl1:ZWINT body derived from Multi-Body 1 (Extended Data Fig. 2c) and the more mobile MIS12C^{Head-1}:MIS12C^{Head-2} body derived from Multi-Body 2 (Extended Data Fig. 2c).

We revised the Methods section to indicate the maps used for modelling and refinement (page 13):

‘The core of the Spc24:Spc25:MIS12C:Knl1 molecules were refined against the consensus map determined at 3.0 Å resolution, whereas the MIS12C^{Head-1}:MIS12C^{Head-2} were refined against Body 2 determined at 3.8 Å (Extended Data Fig. 2c). The composite map of the KMN^{Junction} was reconstructed in RELION 4.0 during MultiBody refinement in which Body 1 and Body 2 were combined on the same coordinates. Model validation was performed against the consensus map determined at 3.0 Å.’

Extended Data Table 1 is revised to indicate the resolution range of 2.7-3.8 Å.

2) The model of the KMN junction shown in Fig 1b is generally expected to be consistent with the cryo-EM map in Fig 1a. I understood that this map was used for structure determination. It is misleading to include the model of long coiled-coil regions which experimental map was not

shown in Fig 1b.

3) The models of coiled-coil regions of Spc24/25 and KNL1/ZWINT were built based on the map with a lower threshold and 2D class averaged images, but the map shown in Fig 2d appears to be too ambiguous and the model was built beyond it. This part of the structure is critical for authors' statement about parallel arrangements of two coiled-coil regions. A map presentation showing model fitting quality is necessary. What is local resolution of the map for these coiled-coil regions? Is it high enough at least to identify each helix? Is it possible to improve the map resolution using the same approach applied to the Mis12C head region? If experimental data is not qualified enough, this structural feature should be described as a hypothetical/predicted model in "Discussion", and then Figs 1d and 1e could be accepted as supportive data.

We thank the reviewer for critically evaluating the structural data that we present. We respond to points 2 and 3 simultaneously:

We performed the MultiBody refinement as suggested by the reviewer. This did indeed improve the coiled-coil densities for both Spc24:Spc25 subunits as well as Knl1:ZWINT. The masks and methods used for data processing of these regions have been added to Extended Data Fig. 3, panel a.

The Spc24:Spc25 coiled-coil is very well resolved. We assigned the register of the chains based on the high-resolution region of the coiled-coil proximal to the RWD domains (also seen in the Fig. 1a), following the register up until the point where coiled-coil can no longer be traced in cryo-EM density. As advised by the reviewer, we truncated the Spc24:Spc25 coiled-coil to only include regions where separate α -helices are observed for both subunits in the cryo-EM density maps.

Similarly, we could observe side chain densities at the bottom of the ZWINT:Knl1 coiled-coil, allowing us to assign amino acids. We traced the coiled-coil in the cryo-EM maps until the density became ambiguous to assign. We also significantly truncated this portion of the Knl1:ZWINT coiled-coil to better correspond to the experimental cryo-EM density.

New pdb models were used in the Fig. 1 to reflect these changes, and new coordinates have been deposited to the PDB. In the Fig. 1b legend we specify the N-terminal residues of Spc24:Spc25^{CC} and Knl1:ZWINT^{CC} fitted to the maps.

4) According to description in Methods (Cryo-EM model building and refinement), AlphaFold2 models of NDC80C and KNL1 were used as initial models, but in Extended Table 1, 'initial model used' is 'ab initio'. Please provide more accurate explanation about initial models used for the structural determination. Was the crystal structure of Mis12C used as an initial model?

Only AlphaFold2 models were used, but manually extensively corrected. Extended Table 1 has been changed to state AlphaFold2 as initial models.

Biochemical analysis and others

1) The authors propose that Dsn1 N-terminus inhibits CENP-C-Mis12C interaction (Fig 2g). To support this proposal, authors performed ITC experiments using CENP-C1-71 and Dsn192-103 peptides to evaluate their binding to Mis12CDsn1 Δ N. However, CENP-C1-71 binds Mis12CDsn1 Δ N much more strongly than Dsn192-103 (Kd of Mis12CDsn1 Δ N -CENP-C1-71 interaction was 0.001 nM, while Kd of Mis12CDsn1 Δ N-Dsn192-103 was 4.92 μ M). This may be caused by using of unappropriated peptides for these ITC experiments, because CENP-C1-71 and Dsn192-103 peptides do not appear to be comparable for their Mis12CDsn1 Δ N binding. In the Mis12C auto-inhibited state structure authors observed, CENP-C6-22 is precluded from Mis12C-binding, while CENP-C24-48 can still interact with Mis12C. To evaluate the auto-inhibition model using ITC, the longer Dsn1 peptide containing both WRR linchpin and RKSL motif and the CENP-C6-22 peptide should be used. Binding of CENP-C6-22 to Mis12CDsn1 Δ N should be compared with that of the longer Dsn1 to Mis12CDsn1 Δ N.

We thank the reviewer for proposing clarifying experiments to test the auto-inhibition model. While addressing the comment, we realized we had a typo in the main text and the Dsn1 peptide used in our work is Dsn1⁹²⁻¹¹³ and not the Dsn1⁹²⁻¹⁰³ as stated. Dsn1⁹²⁻¹¹³, the peptide used in the study, contains both auto-inhibitory elements (both WRR linchpin and RKSL motif), as also proposed by the reviewer. We apologise for this mistake.

As suggested we performed experiments with a peptide modelled on CENP-C residues 2-22 (CENP-C²⁻²²). These data show affinities of CENP-C²⁻²² for MIS12C^{Dsn1 Δ N} and MIS12C^{Dsn1-S100D/S109D} of 10 and 20 nM, respectively, results that are in good agreement with prior data from the Musacchio group using related constructs of MIS12C (Petrovic et al., 2016). The affinity of CENP-C²⁻²² for wild type MIS12C was determined as 485 nM, some 50 and 25-fold lower than to MIS12C^{Dsn1 Δ N} and MIS12C^{Dsn1-S100D/S109D}, respectively, and also in agreement with Petrovic et al., (2016) and Screpanti et al., (2011). Even with the shorter CENP-C peptide, we still observe a substantially higher affinity of CENP-C²⁻²² for all MIS12C variants compared with the Dsn1⁹²⁻¹¹³ peptide. Our findings that CENP-C²⁻²² has a higher affinity than Dsn1⁹²⁻¹¹³ for MIS12C^{Dsn1 Δ N} and MIS12C^{Dsn1-S100D/S109D}, would be consistent with the fact it binds wild type MIS12C with reasonable affinity (0.5 μ M), indicating it can out-compete the Dsn1 auto-inhibitory segment even when the latter interaction with MIS12C is intra-molecular.

2) In Extended Data Fig 5c and d, the authors explain that the auto-inhibition mechanism by Mis12C is likely to be evolutionarily conserved, based on *K. lactis* MIND complex structure predicted by AlphaFold2. However, as the authors mentioned, AlphaFold2 failed to predict the structure of the *S. cerevisiae* MIND complex. Therefore, it is hard to conclude evolutionary

conservation by only the *K. lactis* MIND complex prediction. To strengthen this hypothesis, the authors should provide the sequence alignment of Dsn1 N-terminus across various species and assess whether WRR linchpin and RKSL motif are evolutionarily conserved.

Thank you for this suggestion. A multiple sequence alignment of the DSN1 auto-inhibitory region has been added to Extended Data Fig. 5 (panel e). It shows that both inhibitory motifs are conserved in higher eukaryotes. However, these sequence motifs are absent from many other distantly related species, including yeast. Additionally, the structure of the predicted *K. lactis* Dsn1 auto-inhibitory segment is mainly α -helical, in contrast to the more extended conformation of the human counterpart. This precludes a structure-based sequence alignment. The main text has been modified to reflect this conclusion. Lines 9-32, page 6

3) In the section titled “Dsn1N likely inhibits the binding of the CENP-T to Mis12C”, the authors propose the regulatory mechanisms of the CENP-T-Mis12C interaction, mediated by the Dsn1 N-terminus. This model has already been proposed in previous studies, as the authors mentioned, and no new experimental data about CENP-T-Mis12C interaction is demonstrated in this analysis. This paragraph should be removed, and the essence of the proposal based on the predicted model for the CENP-T-Mis12C complex should be shortly described in the Discussion section.

To our knowledge, this study is the first to present and discuss a model of CENP-T binding to MIS12C, and we can directly compare the CENP-T:MIS12C model prediction with the experimentally observed auto-inhibition by Dsn1, explaining a significant body of prior data, that we hope will be of value to the field. We think that displaying this section more prominently in the main body of the paper will draw more attention to this important comparison. Additionally, it's worth noting that AlphaFold2 only works robustly when the correct peptidic region is isolated and predicted against the folded binding partner. Obtaining a high-confidence AlphaFold2 CENP-T:MIS12C model was not routine and required optimisation. Thus the CENP-T:MIS12C model we report is of scientific value that some readers/researchers might miss out on if they are not advanced users of AlphaFold2.

4) In Extended Data Fig 9, the authors conclude that Knl1mut1 and Knl1mut2 had a modest effect on reducing the affinity of the Knl1-Mis12C interaction. However, in the SEC elution chromatograms (Extended data Fig 9b), Knl1mut2 doesn't appear to affect the Knl1-Mis12C interaction. Please show more convincing data to demonstrate the effect of Knl1mut2. Otherwise, it is hard to conclude that Knl1mut2 reduced binding to Mis12C.

Thank you for highlighting this observation. We agree that Knl1mut2 does not appear to reduce interaction with MIS12C. We have modified the text accordingly:

‘...whereas in contrast, mutating the Knl1^{RWD-C} surface (Knl1^{mut1}: R2248D/W2249A and Knl1^{mut2}: S2270R/S2272W) did not significantly disrupt MIS12C-Knl1 interactions (Extended Data Fig. 9).’ (page 8, lines 33-35).

5) Based on the results from Extended Data Fig 9, the authors conclude that Knl1RWD-N dominates Knl1-Mis12C interactions. This conclusion might not be strictly correct. To disrupt the Knl1RWD-N-Mis12C interaction, Nsl1 C-terminus associated with this interaction was removed. On the other hand, to disrupt the Knl1RWD-C-Mis12C interaction, mutations were introduced in only one of the two important regions of Knl1. Please introduce mutations in both regions involved in the binding with Mis12C in Knl1RWD-C (Knl1R2248D/W2249A/S2270R/S2272W) and investigate the binding with Mis12C, alternatively please change to the correct statement.

Thank you for this suggestion. We generated a combined mutant, and still observed little to no effect on binding to full-length MIS12C. These data have been added to Extended Data Figure 9 and the following text has been added to the manuscript:

‘The Knl1 mutant where the entire Knl1^{RWD-C} surface that interacts with MIS12C has been mutated (Knl1^{mut3}:S2270R/S2272W/R2248A/W2249A) also had little to no effect on the interaction between MIS12C and Knl1 (Extended Data Fig. 9).’ (page 8, lines 35-37).

Minor comments

1) Please show the N- and C-terminus of each protein in Fig 1b for clarity.

We thank the reviewer for this suggestion. We attempted to label the N- and C-termini of each protein in Fig. 1b. However, we found that more than half of termini were obstructed by the protein structures and so difficult to label without hiding or obstructing the structure itself. We also think that adding an additional 16 labels (2 for each protein), made the figure too busy to convey the main message which is the global KMN architecture.

2) In Fig 2a and b, “KSL” should be “RKSL”.

Thank you. This has been replaced with -RKSL-.

3) It is difficult to recognize the hydrophobic pockets of Mis12 and Pmf1 in Fig 2. Please provide an improved structural figure showing the hydrophobic pockets for instance using a surface presentation.

Thank you. New panel Figure 2c with surface representation has been added. The colour scheme has also been changed to be consistent throughout the figure, as requested by referee 1.

4) In Fig 2f, please indicate the locations of His8, Asn11, Arg14, and Arg15 of CENP-C.

His8, Asn11 added to Arg14, Arg15 and Phe17.

5) The deleted region (amino acid residue numbers) in Dsn1ΔN (p5) or Nsl1ΔC (p8) should be described in the main text.

This information has been added to the text on page 5 and page 8.

6) In the graph shown PAE score (Extended Data Fig 5c, Extended Data Fig 6a, Extended Data Fig 8c), please indicate which protein is corresponding to A, B, C, or D in Y-axis.

Protein labels have been added to the Extended Data Fig 5c, Extended Data Fig 6a, Extended Data Fig 8c.

7) In p6 lower part, “MIS12C complex” should be “MIS12C”.

Thank you. This has been corrected.

8) Please correct the numbering of the figure legend of Extended Data Fig7 (d, e, f -> a, b, c)

Thank you. This has been corrected.

9) The head domains in carton schematic in Fig 1c and Fig 5 look like formed by only Pmf1 (green) and Dsn1 (red) only.

This has been corrected in figure 1 and figure 5 to reflect the molecular models.

Decision Letter, first revision:

Message: Our ref: NSMB-A48036A

6th Dec 2023

Dear Dr. Barford,

Thank you for submitting your revised manuscript "Structure of the human outer kinetochore KMN network complex" (NSMB-A48036A). It has now been seen by the original referees and their comments are below. The reviewers find that the paper has improved in revision, and therefore we are happy to accept it in principle in Nature Structural & Molecular Biology, pending relevant textual revisions to satisfy the final requests by reviewer #2 and to comply with our editorial and formatting guidelines.

We are now performing detailed checks on your paper and will send you a checklist detailing our editorial and formatting requirements in about two weeks. Please do not upload the final materials and make any revisions until you receive this additional information from us.

To facilitate our work at this stage, it is important that we have a copy of the main text as a word file. If you could please send along a word version of this file as soon as possible, we would greatly appreciate it; please make sure to copy the NSMB account (cc'ed above).

Sincerely,

Dimitris Typas

Associate Editor
Nature Structural & Molecular Biology
ORCID: 0000-0002-8737-1319

Reviewer #1 (Remarks to the Author):

The authors have done a very good job of responding to my suggestions. The new figure coloring is particularly valuable, and it makes the major "lessons" of the paper much clearer. The Discussion is now also more succinct and to the point.

Reviewer #2 (Remarks to the Author):

This is a revised MS for the structure of KMN network and this reviewer found that authors did additional efforts for revision. Regarding the cryo-EM analysis, the authors have addressed all concerns raised by this reviewer. The revised description of the cryo-EM analysis is notably informative, and the structural presentations have been appropriately enhanced. New biochemical data included in the revised manuscript have strengthened their structural findings and auto-inhibition model.

However, this reviewer still has a slight concern about the section titled "Dsn1N likely inhibits the binding of CENP-T to MIS12C". I generally acknowledge the scientific value of structure predictions using AlphaFold2 (AF2) and understand that the cryo-EM structure of MIS12C solved by the authors allows structural comparison with the predicted MIS12C-CENP-T complex model by AF2. However, to enhance the clarity and precision of the discussion based on the AF2 model, please consider the following point:
On page 7, lines 1-2, the authors described, "AlphaFold2 predicted with high confidence in all models that...". It would be beneficial for the readers if authors could explain why the predicted structure is deemed to have "high confidence." Otherwise, authors can tone down such a description. For instance, in the model displayed in Extended Data Fig 6a, the amino acids of CENP-T are depicted in white or blue, suggesting a reliability is not so high. The reliability of the MIS12C-CENP-T interface could be evaluated by the PAE value (shown in Extended Data Fig 6a as a graph). Is it possible to enlarge/indicate the regions corresponding to MIS12C-CENP-T contact sites in the PAE graph?

Minor comment:

Provide information about the values used for the color-coding in the model presented in Extended Data Fig 6a. pLDDT score, perhaps.

Author Rebuttal, first revision:

Reviewer point-by point 2

Reviewer #1:

Remarks to the Author:

The authors have done a very good job of responding to my suggestions. The new

figure coloring is particularly valuable, and it makes the major "lessons" of the paper much clearer. The Discussion is now also more succinct and to the point.

We thank the reviewer for valuable suggestions that have improved the manuscript.

Reviewer #2:

Remarks to the Author:

This is a revised MS for the structure of KMN network and this reviewer found that authors did additional efforts for revision. Regarding the cryo-EM analysis, the authors have addressed all concerns raised by this reviewer. The revised description of the cryo-EM analysis is notably informative, and the structural presentations have been appropriately enhanced. New biochemical data included in the revised manuscript have strengthened their structural findings and auto-inhibition model.

We thank the reviewer for raising important concerns during the revision of this work. The reviewer's suggestions have significantly improved accuracy and clarity of our work.

However, this reviewer still has a slight concern about the section titled "Dsn1N likely inhibits the binding of CENP-T to MIS12C". I generally acknowledge the scientific value of structure predictions using AlphaFold2 (AF2) and understand that the cryo-EM structure of MIS12C solved by the authors allows structural comparison with the predicted MIS12C-CENP-T complex model by AF2. However, to enhance the clarity and precision of the discussion based on the AF2 model, please consider the following point:

On page 7, lines 1-2, the authors described, "AlphaFold2 predicted with high confidence in all models that...". It would be beneficial for the readers if authors could explain why the predicted structure is deemed to have "high confidence." Otherwise, authors can tone down such a description.

Thank you for this suggestion. We toned down our statement regarding AF2 model: "with high confidence in all models" have been removed from the main text. (top of page 7).

For instance, in the model displayed in Extended Data Fig 6a, the amino acids of CENP-T are depicted in white or blue, suggesting a reliability is not so high. The reliability of the MIS12C-CENP-T interface could be evaluated by the PAE value (shown in Extended Data Fig 6a as a graph). Is it possible to enlarge/indicate the regions corresponding to MIS12C-CENP-T contact sites in the PAE graph?

Extended Data Fig 6a has been modified to highlight the PAE plot region of CENP-T that interacts with MIS12C. This indicates a low PAE score (positional alignment error) for this interaction, and therefore a prediction with high accuracy.

Minor comment:

Provide information about the values used for the color-coding in the model presented in Extended Data Fig 6a. pLDDT score, perhaps.

Thank you for this suggestion. This information has been provided in the Extended Data Fig 6a legend: "The model on the left is coloured by pLDDT score while the model on the right is coloured by chain type" (p31).

Brief summary statement:

Yatskevich et al. determined the structure of the human outer kinetochore KMN network complex. They showed that this forms an extended and rigid rod-like structure and it exists in an auto-inhibited state that can be relieved by phosphorylation.

Final Decision Letter:

Message: 8th Feb 2024

Dear Dr. Barford,

We are now happy to accept your revised paper "Structure of the human outer kinetochore KMN network complex" for publication as an Article in Nature Structural & Molecular Biology.

Your paper will be published online soon after we receive proof corrections and will appear in print in the next available issue. You can find out your date of online publication by contacting the production team shortly after sending your proof corrections.

An online order form for reprints of your paper is available

at <https://www.nature.com/reprints/author-reprints.html>. Please let your coauthors and your institutions' public affairs office know that they are also welcome to order reprints by this method.

Please note that *Nature Structural & Molecular Biology* is a Transformative Journal (TJ). Authors may publish their research with us through the traditional subscription access route or make their paper immediately open access through payment of an article-processing charge (APC). Authors will not be required to make a final decision about access to their article until it has been accepted. Find out more about Transformative Journals

Sincerely,

Dimitris Typas
Associate Editor
Nature Structural & Molecular Biology
ORCID: 0000-0002-8737-1319